# AN OBJECT-CENTRIC SENSITIVITY ANALYSIS OF DEEP LEARNING BASED INSTANCE SEGMENTATION

## ABSTRACT

Understanding if and how instance segmentation models generalize to novel appearances of objects is crucial to many challenging real-world problems such as occlusion or in low-resource domains. In this study we contribute a comprehensive baseline regarding the object-centric robustness of deep learning models. Inspired by work on texture bias in CNNs, we introduce a novel negative test and simulate the out-of-distribution appearance of familiar objects with a stylized version of MS COCO and two counterfactual variants. In addition, we carefully control for potentially confounding corruptions that can lead to disappearing objects in the extreme. The result is a causally well motivated sensitivity analysis. We evaluate a broad range of frameworks including Cascade and Mask R-CNN, Swin Transformer, BMask, YOLACT(++), DETR, BCNet, SOTR and SOLOv2. We find that YOLACT(++), SOTR and SOLOv2 are significantly more robust to corruptions and novel texture compared to competitive models ($\sim 10 - 30\%$). In general, we show that deeper and dynamic architectures improve robustness whereas training schedules, data augmentation and pre-training have only a minor impact. In summary we evaluate 68 models on 61 versions of COCO for a total of 4148 evaluations.

## 1 INTRODUCTION

In this study, we want to learn how deep learning models for instance segmentation generalize to novel appearances of familiar objects. Despite their remarkable success in computer vision, deep neural networks still struggle in challenging real-world scenarios (Yuille & Liu, 2018; Michaelis et al., 2019; Recht et al., 2019; Madan et al., 2021). Consider for instance a pedestrian with an unconventionally textured dress or a rare horse statue made from bronze. The model might have seen many pedestrians or natural horses during training but still fails to detect these unseen and out of context examples, often with high confidence. Generalizing to such naturally adversarial objects is typically described as **out-of-distribution robustness** in the literature (Hendrycks et al., 2021; Lau et al., 2021). Interestingly, Hendrycks et al. (2021) suggest that improvements in this direction are more likely to come from computer vision architectures than from existing data augmentation or additional public datasets. With the ever increasing number of competitive models, our objective is therefore to contribute an extensive comparison of instance segmentation models to unveil promising future research directions. The perspective we take is inspired by the work of Baker et al. (2018) and Geirhos et al. (2019) on texture bias in classification models. More precisely, both groups found that when compared to humans, convolutional neural networks (CNNs) mostly ignore object shapes, i.e. ignore the defining structure of an object. In fact, Brendel & Bethge (2019) have further shown that CNNs can robustly classify objects in texturized images where the global appearance of objects is fully mixed up. In summary, these findings indicate that a revision of contemporary deep learning architectures is a much needed contribution on the pathway to more systematic generalization. To compare the robustness of instance segmentation models, we introduce a novel negative test in the form of an object-centric sensitivity analysis. More precisely, we investigate the impact of increasingly novel **object texture** while controlling for the effect of **corrupted color**, **shape** and **contour** features. By object-centric, we mean that the appearance of visual objects is changed in a semantics preserving and causally plausible way, i.e. is controlled at the instance level.

Since we generally expect that segmentation performance will degrade under novel textures and increasing corruptions, our benchmark can be understood as a negative test, i.e. if a specific model

Figure 1: Left: Simplified creation process of the Stylized COCO dataset. Style images are randomly chosen from Kaggles Painter by Numbers dataset. Right: We use mask annotations to create counterfactual, object-centric versions of Stylized COCO. We append more examples of the creation process in the Appendix.

appears to be significantly more robust than others, we can consider it a promising candidate for more in depth research on generalization. To simulate novel appearances of familiar objects we utilize a stylized version of MS COCO (Lin et al., 2014) as shown Figure 1 left. As can be seen, the AdaIN method (Huang & Belongie, 2017) effectively replaces local texture cues but preserves the global shape of objects. In addition, we create two object-centric versions of the stylized COCO dataset as displayed in Figure 1 right. By comparing the performance on all three datasets we can control for spurious correlations that might have been introduced in the fully stylized dataset. Note that the style masking encodes ground truth information from the test set in the data which could potentially be exploited by the models. While this would be an interesting finding in itself, we do not see this to happen in practice. A limitation of our approach is the use of artistic style images. This leads to novel object appearances that can not be found in real-life directly. However, we compromise on this choice since it ensures that textures are truly novel and not biased which could happen with natural style images. We discuss more alternatives to our simulation method in the related work section.

A closely related perspective on segmentation robustness can be found in work that acknowledges the inherent **long-tail distribution** of real-world data. In such settings, the challenge is to become robust against the bias of extreme class imbalance in existing datsets. Common approaches to resolve this issue are re-sampling (Wang et al., 2020b; Chang et al., 2021) and regularization strategies (Tang et al., 2020; Pan et al., 2021; Hsieh et al., 2021; Wang et al., 2021b). The long-tail problem can also be understood as a low resource setting where data collection is expensive or class labels are missing. In such cases, the challenge is to efficiently adapt to novel objects or uncommon appearances in a semi-supervised manner (Hu et al., 2018b; Fan et al., 2020). A particular instantiation of this problem is object occlusion. Since objects can occlude each other in almost infinitely different ways, a common strategy is to model object features more explicitly, i.e. to decouple shape and appearance for instance (Chen et al., 2015; Cheng et al., 2020; Ke et al., 2021; Fan et al., 2020). The idea is to learn representations that generalize more systematically and we expect these methods to be strong contenders in our comparison. We hope that our comprehensive benchmark motivates more research in this direction which we believe, will lead to improvements in all of the related problem spaces.

## 2 METHODS

In this section we describe the datasets, frameworks and models that are used in this study. The code to reproduce our results, as well as $\approx 1.5$TB of detection and evaluation data can be found here: `link-to-project-page-when-published`.

### 2.1 AN OBJECT-CENTRIC CAUSAL VERSION OF STYLIZED COCO

Stylized COCO as shown in Figure 1 left is an adaptation of Stylized-ImageNet by Geirhos et al. (2019). It was first used by Michaelis et al. (2019)[1] as data augmentation technique during training to improve robustness of detection models against common corruption types such as gaussian noise or motion blur. We instead use a stylized version of the `val2017` subset to test instance segmentation models on this data directly. By manual inspection of Stylized COCO, we found that strong

---

[1] Stylized Datasets: `https://github.com/bethgelab/stylize-datasets`

stylization can sometimes lead to images where the object contour starts to vanish, up to the point where objects and their boundaries dissolve completely. This effect depends on the style image used in the creation process but affects objects of all scales alike. As shown in Figure 2, we resolve this issue by using the ground truth mask annotations to limit the style transfer to the actual objects or the background. This not only ensures that object contours are preserved but also controls for global stylization as a confounding variable. By assuming an object-centric causal model, Stylized COCO allows us to ask interventional questions regarding the original COCO dataset, e.g. *"What happens if we change the texture of images?"*. By masking the style transfer to objects or background, we can also ask counterfactual questions such as *"Was it actually the object that caused the change in performance?"*, *"What if we change the background instead?"*. We will refer to the different dataset versions as Stylized COCO (●), Stylized Objects (▲) and Stylized Background (■).

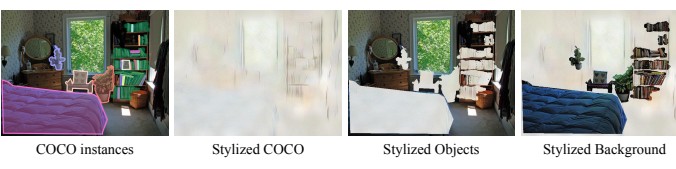

COCO instances    Stylized COCO    Stylized Objects    Stylized Background

Figure 2: Depending on the style image, object boundaries can vanish due to strong stylization. The Stylized Objects and Background versions of Stylized COCO resolve this issue.

A related, second problem that remains after masking is that shape information within the object can also be lost due to strong stylization. We address this issue by controlling the strength of the AdaIN style transfer. This can be done with an $\alpha$ parameter that acts as a mixing coefficient between the content and style image. More precisely, AdaIN employs a pre-trained VGG encoder $f$ on both images, performs an interpolation step between the resulting feature maps and produces the final output with a learned decoder network $g$. In summary, a stylized image $t$ is produced by $T(c, s, \alpha) = g((1 - \alpha)f(c) + \alpha \text{AdaIN}(f(c), f(s)))$ where $c$ and $s$ are the content and style images respectively. We will refer to this method as blending in *feature space*. The top row of Figure 3 shows two examples of the extreme points $\alpha = 0$ and $\alpha = 1$. Note that at $\alpha = 0$, the image colors are mostly preserved but the algorithm has already introduced artifacts in the form of subtle texture and shape changes. In response we create a control group where we perform alpha blending between the pixel values of the original content image $c$ and the fully stylized image $t_{\alpha=1}$: $P(c, t_{\alpha=1}, \alpha) = ((1 - \alpha) * c + \alpha * t_{\alpha=1})$. We will refer to this method as blending in *pixel space*. In contrast to the feature space sequence, the control group should preserve textures and object shape over a longer range. The idea is to compare models on both sequences in order to attribute performance changes to either color and texture or shape, depending on the objects size. In contrast to Geirhos et al. (2019) who used a fixed style strength to modify ImageNet features ($\alpha = 1$), we produce the full alpha-range $\alpha \in (0.0, 0.1, 0.2, ..., 1.0)$ for both blending spaces. Note that every alpha value depicts a separate and complete version of the accordingly styled COCO `val2017` subset. The qualitative differences can be inspected in Figure 3 bottom left (zoom in for better visibility).

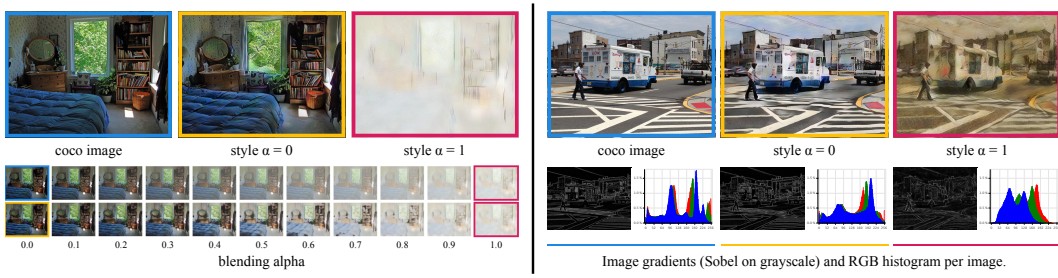

coco image    style α = 0    style α = 1        coco image    style α = 0    style α = 1

0.0  0.1  0.2  0.3  0.4  0.5  0.6  0.7  0.8  0.9  1.0
blending alpha        Image gradients (Sobel on grayscale) and RGB histogram per image.

Figure 3: Top row: Comparison of COCO and Stylized COCO at different alphas. The AdaIN method introduces subtle artifacts even at $\alpha = 0$ (no style). Bottom left: We control the style strength in feature space (yellow to pink) and pixel space (blue to pink). Note that every alpha value in these sequences depicts a separate and complete version of the accordingly styled COCO `val2017` subset. Bottom right: Comparison of image gradients and color histograms at different alphas.

We also went for quantitive measures to validate our subjective impression of Stylized COCO. Figure 3 bottom right shows a comparison of image gradients and RGB histograms at the *extreme* points. Compared to the original image we can see the subtle shape changes in the gradient map of $\alpha = 0$ and a significantly different color histogram at $\alpha = 1$. To describe this effect over the full

alpha range, we compute the structural similarity index (SSIM) (Wang et al., 2004) between gradient images of corresponding image pairs. Between RGB histograms we compute the Wasserstein distance alike. We always compare against the original COCO data and report the mean distance averaged over the full dataset at a specific alpha. In addition to the image-to-image scores we also include an instance level comparison for the COCO scales S,M and L. Instances have been cropped based on bbox information. This addition was added after we observed that small objects appear to be more affected by the AdaIN artifacts compared to medium and large instances. Figure 4 displays the results and confirms our assumption. Structural similarity depends on object size and is in fact, almost constant over the full feature space range for small objects. Furthermore, the control group preserves structural similarity over a longer range as intended. Color distance in contrast converges at around $\alpha = 0.3$. Based on these insights, we feel confident to better attribute potential performance dips and subsequently, determine the relative importance of each feature type.

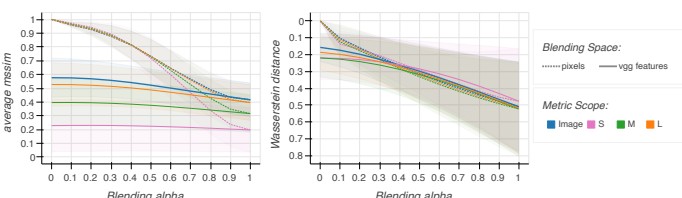

Figure 4: Left: Average structural similarity between image gradients in relation to COCO (a score of 1 means that there is no difference between images). Right: Wasserstein distance between RGB histograms (reversed y-axis).

## 2.2 MODEL SELECTION

To contribute a comprehensive overview on model robustness, we opted for a broad comparison of popular frameworks and architectures. The dimensions we consider to be impactful are **framework**, **architecture** and **pre-training**. The selected models can be found in Table 1.

**Frameworks** for instance segmentation can be categorized in different ways. A first distinction can be made between methods that solve the detection problem as a refinement process of box proposals (multi stage) and methods that predict bounding boxes directly (one stage). We include the popular multi-stage frameworks Mask R-CNN (He et al., 2017) and Cascade Mask R-CNN (Cai & Vasconcelos, 2018) that uses multiple refinement stages instead of one. Both frameworks formulate instance segmentation as a pixel-wise classification problem. Since this rather naive extension to Faster R-CNN (Ren et al., 2015) can ignore object boundaries and shapes, we include the boundary-preserving mask head alternative (Cascade-) BMask (Cheng et al., 2020) for comparison. A remaining challenge to boundary detection are overlapping objects that occlude the ground truth contour of other instances. We therefore include the Bilayer Convolutional Network (BCNet) (Ke et al., 2021) as another mask head alternative. In BCNet, the occluded and occluding object are separately detected and modeled explicitly in a layered representation. The mask head can then "consider the interaction between [the decoupled boundaries] during mask regression (Ke et al., 2021)." A second distinction between frameworks concerns the use of predefined anchor boxes. Anchor based methods predict relative transformations on these priors whereas anchor free methods predict absolute bounding boxes. We include YOLACT(++) (Bolya et al., 2019; 2020) as a one-stage, anchor based framework. YOLACT is a real-time method that solves instance segmentation without explicit localization (feature pooling). Instead, it generates prototype masks over the entire image which are combined with per-instance mask coefficients to form the final output. The (++) version improves by adding a mask re-scoring branch (Huang et al., 2019a) and deformable convolutions (v2) (Dai et al., 2017; Zhu et al., 2019). We include DETR (Carion et al., 2020) as a one-stage, anchor free framework that formulates object detection as a set prediction problem over image features. Note that it was not primarily designed for instance segmentation but offers a corresponding extension that we use in our study. Based on model availability we include BCNet in the FCOS (Tian et al., 2019) variant (F-BCNet). FCOS is a fully convolutional, one-stage, anchor-free alternative to Faster R-CNN that "solves object detection in a per-pixel prediction fashion, analogue to semantic segmentation (Tian et al., 2019)." Finally, we distinguish between top down frameworks where detection precedes segmentation and bottom up methods where bounding boxes are derived from mask predictions. We include the bottom-up methods SOLOv2 (Wang et al., 2020d) and SOTR (Guo et al., 2021). SOLO (Wang et al., 2020c) divides the input into a fixed grid and predicts a semantic category and corresponding instance mask at each location. The final segmentation is obtained with non-maximum-suppression on the gathered grid results to resolve similar predictions of adjacent

grid cells. SOLOv2 improves by introducing dynamic convolutions to the mask prediction branch, i.e. an additional input dependent branch that dynamically predicts the convolution kernel weights. A similar idea was used by Tian et al. (2020b). SOTR uses a twin attention mechanism (Huang et al., 2019b) to model global and semantic dependencies between encoded image patches. The final result is obtained by patch wise classification and a multi-level upsampling module with dynamic convolution kernels for mask predictions, similar to SOLOv2. For completeness, we also include YOLO(v3,4 and scaled v4) to our comparison since detection is a vital sub-task of top down frameworks (Redmon & Farhadi, 2018; Bochkovskiy et al., 2020; Wang et al., 2021a).

Table 1: Overview of frameworks, backbones and neck methods. (*) Swin Transformer use hierarchical representations similar to FPN necks in CNNs. RegNetY is similar to RegNetX but implements the Squeeze-and-Excitation operation (Hu et al., 2018a). Yolo consists of darknet (D), spatial pyramid pooling (SSP) (He et al., 2014) and a Path Aggregation Network (PAN) (Liu et al., 2018) in varying combinations with CSPNet (C) (Wang et al., 2020a).

| Backbone | Framework | | | | | | | | |
|---|---|---|---|---|---|---|---|---|---|
| CNN | multi stage | | | one stage | | | | | |
| GCN | anchor based | | | | | | anchor free | | |
| Hybrid | top down (bbox→segm) | | | | | | | bottom up (segm→bbox) | |
| ViT | Mask R-CNN | BMask | Cascade | YOLO(v3,4,s4) | YOLACT(++) | DETR | FCOS BCNet | SOTR | SOLOv2 |
| R50 | FPN, C4, DC5, DCN | FPN | FPN | - | FPN, DCN | FPN, DC5 | - | - | FPN |
| R101 | FPN, C4, DC5, | FPN | DCN | - | FPN, DCN | FPN | FPN | FPN, DCN | FPN |
| X101 | FPN | - | - | - | - | - | - | - | - |
| X151 | - | - | FPN, DCN | - | - | - | - | - | - |
| RegNetX | FPN | - | - | - | - | - | - | - | - |
| RegNetY | FPN | - | - | - | - | - | - | - | - |
| Swin-T | FPN* | - | FPN* | - | - | - | - | - | - |
| Swin-S | FPN* | - | FPN* | - | - | - | - | - | - |
| Swin-B | - | - | FPN* | - | - | - | - | - | - |
| D53 | - | - | - | FPN | - | - | - | - | - |
| CD53 | - | - | - | (C)PAN, SPP | - | - | - | - | - |

**Architectures** used in instance segmentation can be divided into backbone, neck and functional heads. The latter output the final results and are framework specific. Backbones and necks however are typically chosen from a pool of established models which allows for a controlled comparison. The role of backbone networks is to extract meaningful feature representations from the input, i.e. to encode the input. The neck modules define which representations are available to the functional heads, i.e. define the information flow. We include the CNN backbones ResNet (He et al., 2016), ResNext (Xie et al., 2017) and RegNet (Radosavovic et al., 2020), a network found with meta architecture search that outperforms EfficientNet (Tan & Le, 2019). Note that BCNet utilizes a Graph Convolutional Network (GCN) (Kipf & Welling, 2017) within its mask heads to model long-range dependencies between pixels (to evade local occlusion). Furthermore, DETR and SOTR are hybrid frameworks that use transformer architectures to process the encoded backbone features. With Swin Transformer (Liu et al., 2021) we also include a convolution free backbone alternative based on the Vision Transformer approach (ViT) (Dosovitskiy et al., 2021). The most popular neck choice is the Feature Pyramid Network (FPN) (Lin et al., 2017). It builds a hierarchical feature representation from intermediate layers to improve performance at different scales, e.g. small objects. For comparison we also include a ResNet conv4 neck (C4) as used in Ren et al. (2015) and a ResNet conv5 neck with dilated convolution (DC5) as used by Dai et al. (2017). Finally we abbreviate FPN models that use deformable convolutions as DCN (Dai et al., 2017; Zhu et al., 2019). Similar to dynamic convolutions which predict kernel weights, DCNs learn to dynamically transform the sampling location of the otherwise fixed convolution filters.

**Pre-training** of backbone networks is commonly done as supervised learning on ImageNet (IN). Due to the recent success of self supervised learning (SSL) in classification, we are interested in how these representations perform in terms of object-centric robustness. In particular we are interested in the contrastive learning framework that seeks to learn "representations with enough invariance to be robust to inconsequential variations (Tian et al., 2020a)". Based on availability we include the methods InstDis (Wu et al., 2018), MoCo (He et al., 2020; Chen et al., 2020), PIRL (Misra & van der Maaten, 2020) and InfoMin (Tian et al., 2020a). Note that pre-trained backbones were only used as initialization for a supervised training on COCO. As a final comparison we include models that have been trained with random initialization and Large Scale Jittering (LSJ) (Ghiasi et al., 2021) data augmentation as an alternative to pre-training.

The complete list of models can be inspected in Figure 5. From the overview in Table 1 we can see that our selection allows for a fair comparison of frameworks as long as we fix the backbone and neck architecture to ResNet+FPN. Vice versa we can compare backbone and neck combinations within a fixed framework, in particular Mask R-CNN. Note that we also compared different learning

schedules but did not include this dimension more prominent in the categorization after finding it to be the least significant factor in our evaluation.

# 3 RELATED WORK

The classical understanding of robustness is concerned with corruptions that stem from signal processing errors such as gaussian or salt and pepper noise etc. (Hendrycks & Dietterich, 2019; Michaelis et al., 2019; Kamann & Rother, 2021; Mummadi et al., 2021). A popular alternative is to understand robustness by comparing model performance and failure cases against humans (Geirhos et al., 2018; 2019; 2020; 2021; Shankar et al., 2020; Tuli et al., 2021). Geirhos et al. (2021) and Tuli et al. (2021) for instance find that Transformer models perform more consistent to humans than CNNs. Madan et al. (2021) on the other hand report that both model types are prone to small in-distribution changes in 3D perspective and lighting. We contribute to this matter by including both architectures in our comparison. A third perspective is given by work that aims to compensate for the long-tail distribution in real-world data. In this case, robustness can be understood as the ability to adapt to novel object classes or uncommon appearances in a data efficient transfer learning process (Hu et al., 2018b; Fan et al., 2020). The approach we take is more direct and measures robustness in a challenging zero-shot generalization setting. As an alternative to our sensitivity benchmark, Islam et al. (2021) analyze feature importance in latent representations and Cao & Johnson (2021) use feature visualization to understand object detectors. Both leverage style transfer to simulate novel object appearances. The closest real-life alternative is the Natural Adversarial Objects (NAO) dataset (Lau et al., 2021). It poses a more realistic long-tail setting but does not allow to control for pose and perspective, i.e. to observe the exact same object with varying textures for instance. Other alternatives are the 3DB framework (Leclerc et al., 2021), a rendering engine that enables artifact free texture transfer on synthetic objects and SI-Score (Yung et al., 2021), a dataset for analyzing robustness to rotation, location and size.

# 4 RESULTS

In this section we present the evaluation results on Stylized COCO, Objects and Background. Each dataset version contains 20 copies of the accordingly styled COCO `val2017` subset. As a reference point, we first report the absolute Average Precision (AP) on the original `val2017` subset for all models in Figure 5. As can be seen, training schedule, data augmentation and architecture choice have the biggest impact within a framework. Overall, RegNets trained with LSJ and Swin Transformer models perform best. Note that SOTR and SOLO have worse APs but significantly better APm and APl compared to other frameworks (see Appendix).

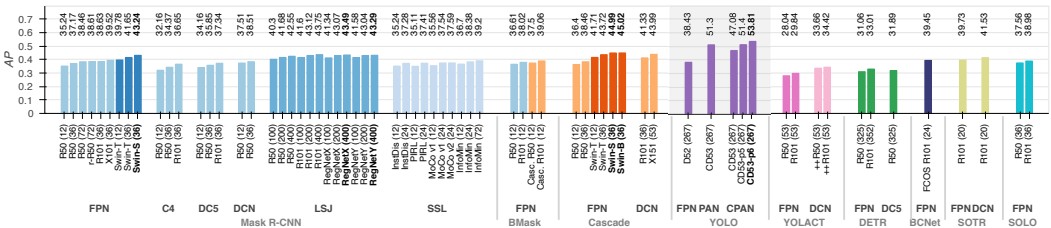

Figure 5: Absolute performances on COCO `val2017`. Training schedules in epochs have been appended to model names. Note that Yolo is bbox AP which is not directly comparable but included for model completeness. Methods that do not report scores for `val2017` have been validated on `test-dev2017` first.

We now present the results of our object-centric sensitivity analysis. We first provide a general overview of the full blending sequences and then focus on the *extreme* points $\alpha = 0$ and $\alpha = 1$ in more detailed comparisons. In total we tested 68 pre-trained models [2] on 61 datasets which sums up to 4148 subset evaluations. To select models for grouping in the overview and reporting in the detailed comparisons, we calculated the distance matrix between all models for each dataset. These can be found in the Appendix together with more metrics and the corresponding bounding box scores.

---

[2] See Appendix for the complete list of code projects and weight sources.

Note that we focus primarily on the AP score for segmentation and the scale dependent COCO metrics APs, APm and APl due to our object-centric perspective. To quantify object-centric robustness, we report the relative performance in comparison to the uncorrupted data. For every dataset version, blending space and alpha value we calculate: $rP_\alpha = P_\alpha/P_{coco}; P \in \{AP, APs, APm, APl\}$. Note that we release the exact numerical values together with our code since reporting this amount of data in the form of tables is not expedient.

Figure 6 displays **the average relative performance per framework** and stylized COCO version for the *feature space blending sequence* (dark) and the *pixel space control group* (light, always starting from 100%). We can immediately see that YOLACT(++) is consistently more robust (rAP) than all other frameworks due to its strong performance on small objects (rAPs). Similarly, the bottom-up frameworks SOTR and SOLOv2 are significantly more robust on medium (rAPm) and large objects (rAPl). Within the Cascade and Mask R-CNN frameworks, Swin Transformer are slightly more robust than their CNN based counterparts. To our surprise, the advanced BMask and BCNet models do not consistently improve robustness but rather closely follow the performance of the naive mask head designs. We investigate this not expectable ranking in more detail in Figure 7 and provide some further explanation in our discussion.

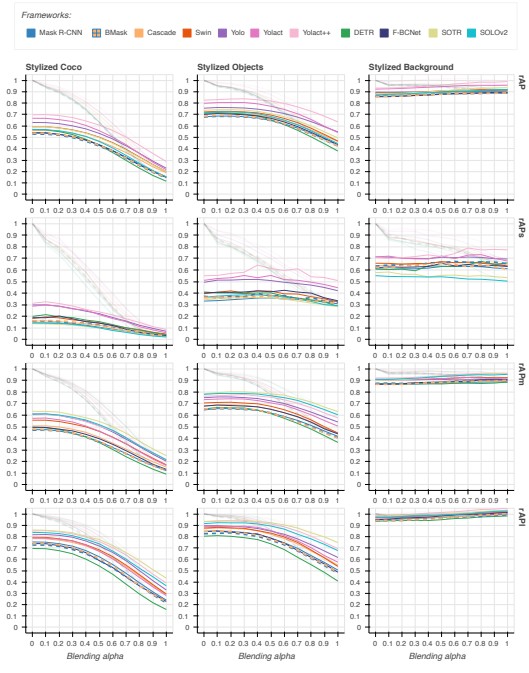

Figure 6: General overview of model robustness (zoom in for better visibility). Masking the style transfer effect to ground truth objects results in improved relative performance without changing the overall trend. Structural similarity as displayed in Figure 4 seems highly predictive for the low alpha range but can not explain the further decreasing robustness values. We also like to point out the small performance dip of the control group at $\alpha = 0.1$ that seems to be induced by color shift. Surprisingly, increasing the style strength on background features appears to have a positive effect for large objects (rAPl). Swin Transformer are display as a separate group due to their convolution free backbone architecture. Yolo is bounding box rAP which is not directly comparable but included for model completeness. Note that absolute metrics result in different ordering due to varying base performance. We append this version together with the corresponding bbox variants in the Appendix.

Note that our object-centric sensitivity approach allows to validate these findings in a causally rigorous way for the first time. More precisely, it allows us to distinguish between the impact of stylization artifacts (artificial signal noise) and novel object appearances (out of distribution texture). This is not possible with only single style strength tests as typically done. In general, it can be expected that all models will degrade in performance with increasing style strength at some point. We can see however that after an initial performance loss at $\alpha = 0$ (subtle corruptions of original shapes and textures), models are fairly robust until actual out of distribution texture from the style image is introduced (starting around $\alpha = 0.4$). This effect applies to medium and large but not small objects where the performance appears to be fully dominated by the artifacts of the AdaIN method. As a result, we can respect these insights in our conclusions. For instance, we now understand that for medium and large objects, texture is more important to instance segmentation models than object shape or subtle corruptions. From the gap between the *pixel space* control group and the *feature space* blending sequence we can further tell that subtle shape and texture corruptions are more severe than color changes. In addition, we can validate that our findings can be attributed to actual objects by comparing the performance on Stylized Objects and Background. In general, models are heavily relying on context information for small objects but not for medium and large instances. Again, this could be assumed but not be proven with the scientific evidence of only one, fully stylized dataset.

Finally, we like to point out that models perform significantly better on Stylized Objects than on the fully Stylized COCO. Within the limitation of our data creation process, we conclude that instance segmentation models do indeed consider object contour as a predictive feature.

For a **fair comparison between frameworks** we fix the backbone and neck to ResNet-50 and FPN. Figure 7 displays the results from the feature blending sequence at $\alpha = 0$ and $\alpha = 1$. We can again observe that object-centric stylization is important but does not change the general ranking. Similarly, small objects are more directly affected by shape and texture artifacts whereas medium and large objects degrade due to texture transfer. Between Cascade and Mask R-CNN as well as BMask and BCNet we find only minor differences. YOLACT(++) performs consistently better across all scales and SOLO and SOTR perform on par or even better but struggle with small objects. DETR shows the opposite behavior.

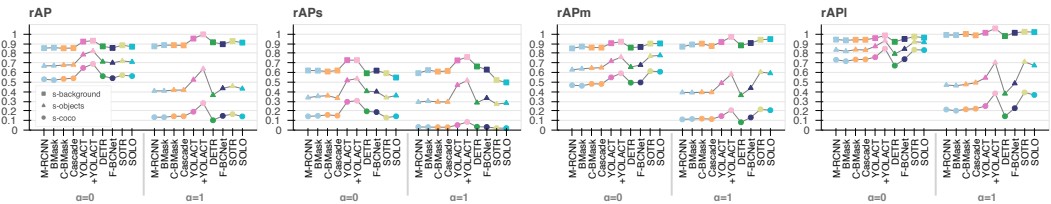

Figure 7: Object-centric robustness by framework. Note that we compromised on R-101 for SOTR and F-BCNet, see Table 1 for an overview of available backbone combinations.

To investigate **the impact of different backbones** we fix the neck method (FPN) and compare model pairs within a given framework. As can be seen from the rAP metric in Figure 8 (top), deeper models perform generally slightly better. The rAPm and rAPl metrics follow this trend. We therefore append the results for rAPs were the behavior is less clear and surprisingly, often reversed. Note that Swin-T and S are comparable to ResNet-50 and 101 in model complexity and perform best within the Cascade and Mask R-CNN framework respectively. BM and C-BM represent BMask and Cascade BMask respectively.

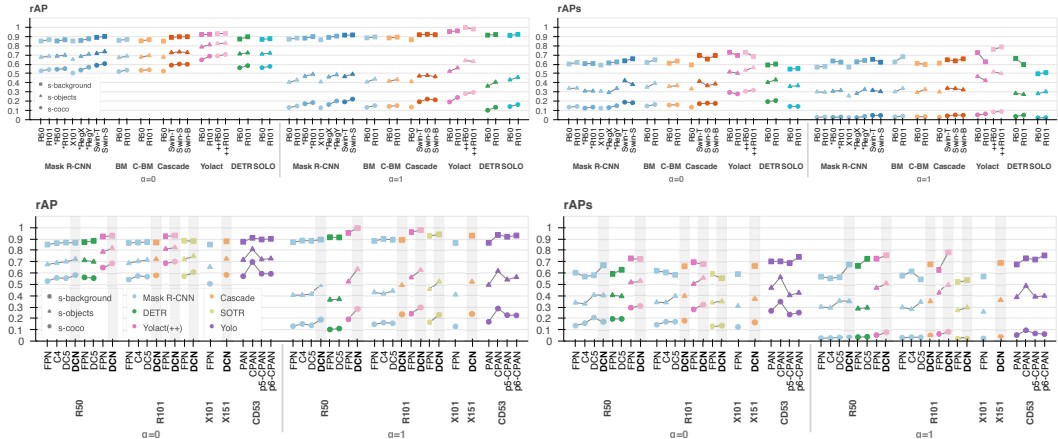

Figure 8: *Top:* Object-centric robustness by backbone architecture (zoom in for better visibility). Models marked with * are trained with LSJ. *Bottom:* Robustness by neck architecture. Deformable convolutions are highlighted for better visibility. Note that Yolo scores are bbox and not directly comparable.

To investigate **the impact of different neck methods** we do the reverse and compare models by fixing backbones and framework. As can be seen from Figure 8 (bottom), FPN necks are typically the least robust option. However, FPN in combination with deformable convolutions (DCN) performs consistently best. Note that SOTR additionally implements dynamic convolution kernels. For small objects, DC5 necks improve the robustness on Stylized Objects (triangle). We also compare Yolo against itself and find that surprisingly, the simple v4-csp model (CPAN) is the most robust option. We provide hypothesis for these findings in our discussion.

Finally we investigate **the impact of pre-training** and strong data augmentation. As displayed in Figure 9, different types of initialization perform almost similar within a few percent. This is surprising since we expected representations to perform more differently based on the objective used in pre-training, in particular from the contrastive learning framework. On the other hand we can report that supervised and unsupervised representations perform almost on par. Similarly, longer training schedules have little impact on model robustness but do not lead to strong overgeneralization either. In comparison we find that random initialization with LSJ data augmentation performs best.

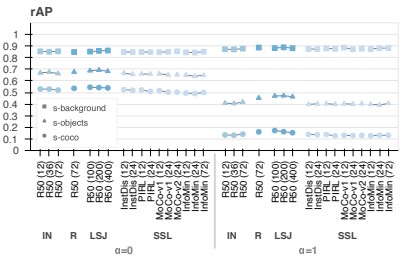

Figure 9: Object-centric robustness by pre-training and data augmentation.

## 5 DISCUSSION

The first key finding of our object-centric sensitivity analysis is that YOLACT and the bottom-up frameworks SOTR and SOLOv2 are significantly more robust than other frameworks. We suspect that the consideration of the entire image in combination with their mechanisms to represent multiple *mask prototypes* to play a major role, though more in depth research is required to verify this hypothesis. To our surprise, the object specific mask heads BMask and BCNet did not improve robustness in a significant way. A naive explanation might be that their specialized representations do simply not generalize to out-of-distribution texture. However, it could also be the case that improved mask detections are just not well reflected in the averaged AP scores. Since we release all detection data, this hypothesis could be investigated by extensive result visualization in a follow-up work. In general we find that deep learning models for instance segmentation appear to be biased towards texture, similar to classification models (Geirhos et al., 2019). More specifically, we show that models are fairly robust to shape and texture corruptions after an initial performance dip. The remaining loss can then be attributed to actual texture transfer from an unknown domain. This effect appears to scale with instance size and we hypothesize that the difference between shape and texture simply collapses for (very) small objects in MS COCO. In contrast to Geirhos et al. (2019) who argue that this problem is induced by the ImageNet training data, we find no such correlation in terms of pre-training. We conclude that either a similar bias is induced by the COCO dataset or that architectures are simply not flexible enough to cope with feature variability, independent of training regime. The latter hypothesis was formulated by Greff et al. (2020) as **the binding problem** and concerns "the inability [of neural networks] to dynamically and flexibly combine (bind) information [... which] limits their ability to [...] accommodate different patterns of generalization". Our comparison of fixed and dynamic architectures supports this hypothesis as we find that deformable and dynamic convolutions lead to improved robustness consistently. We consider this another promising research direction towards systematic generalization.

## 6 CONCLUSION

In this study we contribute a comprehensive baseline on the object-centric robustness of instance segmentation models. We first introduce a novel negative test on Stylized COCO in the form of an counterfactual sensitivity analysis. Based on a broad selection of frameworks and architectures, we test 68 models on 61 dataset versions for a total of 4148 subset evaluations. We find a non-trivial ranking of frameworks with YOLACT(++), SOTR and SOLOv2 performing best. In addition we show that dynamic and deeper architectures improve robustness. In contrast, training schedules, data augmentation and pre-training have only a minor impact in comparison. We discuss limitations of our approach and provide interesting future research directions in our discussion.

REPRODUCIBILITY STATEMENT

We want to thank the authors of Detectron 2, BMask, PyContrast, Swin, YOLO, YOLACT(++), DETR, BCNet, SOTR and SOLOv2 for their contribution of publicly available model weights. Without their commitment to open research, this study would not have been possible. We follow their example and publish all code and data used in this study to ensure reproducibility and more research moving forward.

The code to reproduce our results on Stylized COCO, Stylized Objects and Stylized Background can be found here: `link-to-project-page-when-published`. We also provide $\approx 1.5\text{TB}$ of detection and evaluation data. We include the latter to minimize GPU availability as a limiting factor for future research. To improve actual reproducibility, we also provide automated environment setups for the dataset creation and each of the instance segmentation frameworks.

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

# APPENDIX

## DATASET CREATION

Figure 10 provides a representative sample of images in Stylized COCO. Figure 11 displays the effect of controlling the style strength parameter in AdaIN. Figure 12 shows the masked versions of the last example image in Figure 11 and its blending sequences for Stylized COCO. Finally we attached a complete example for one image in Figure 13.

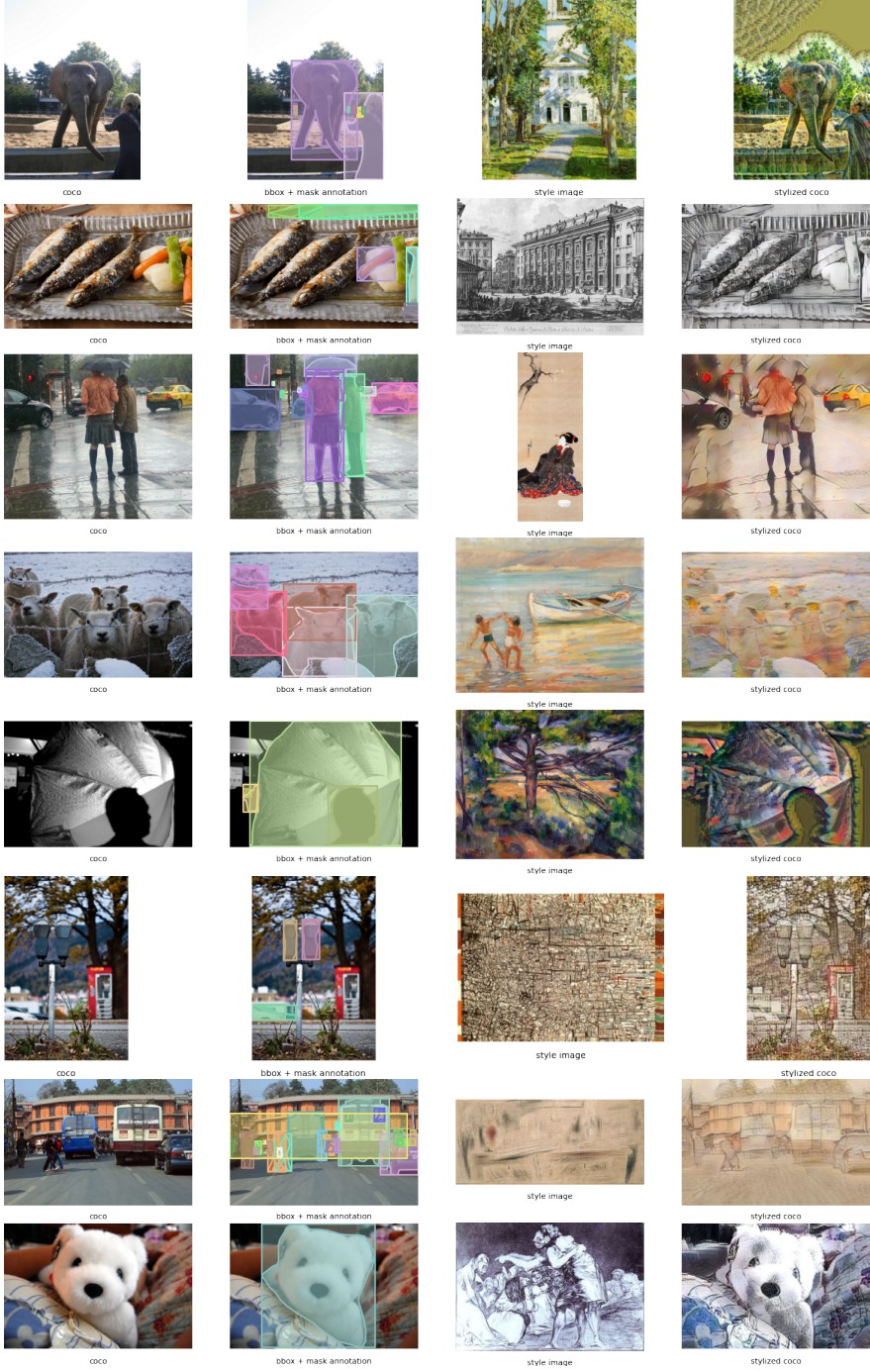

Figure 10: Creation process of Stylized COCO. We plot the mask annotations to locate ground truth instance in the stylized images. These are also used to create the masked version of Stylized COCO.

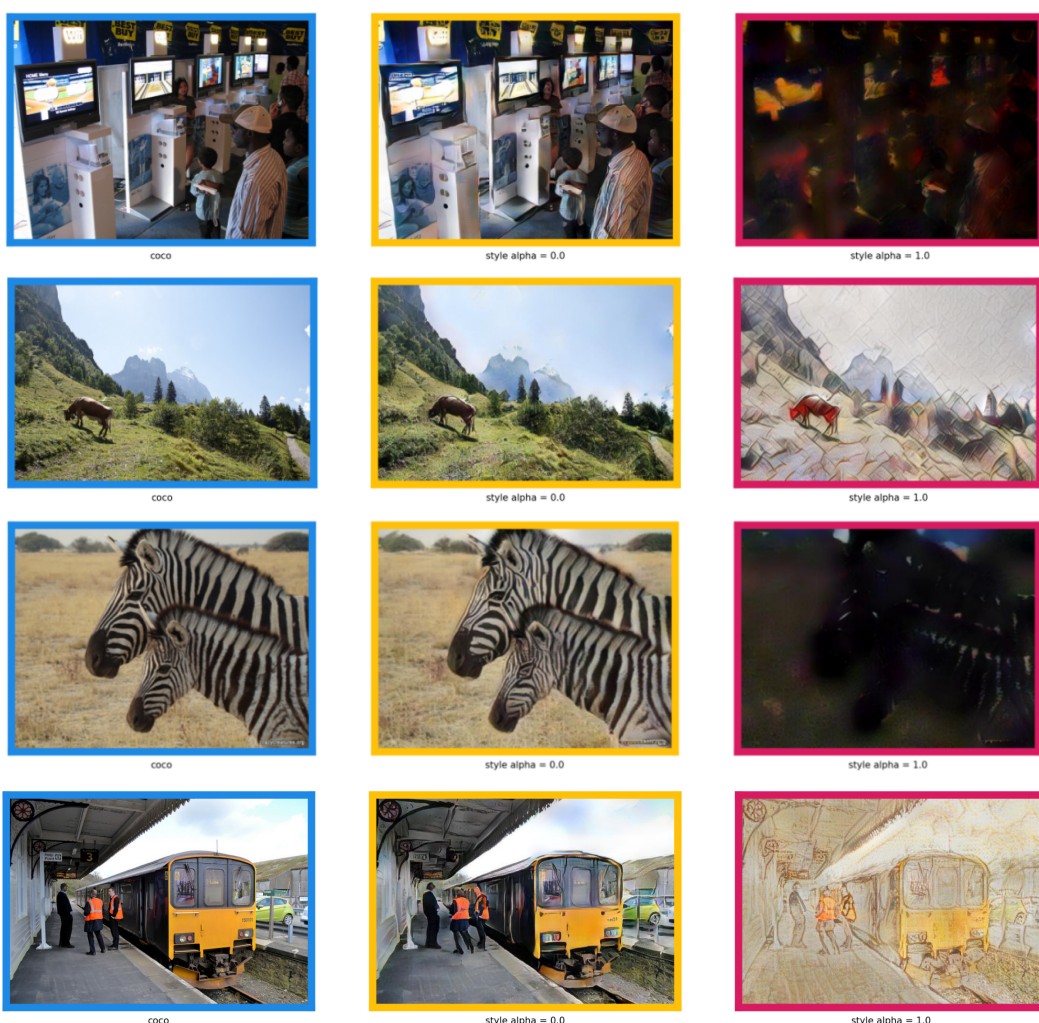

Figure 11: Comparison of AdaIn style transfer strength. Depending on the style image, an alpha value of 1 (pink) can produce rather extreme versions where objects are almost eradicated from the scene. An alpha value of 0 (yellow) corresponds to a style transfer of the content image with itself. As can bee seen in the middle column, this variant already introduces subtle shape changes to the images.

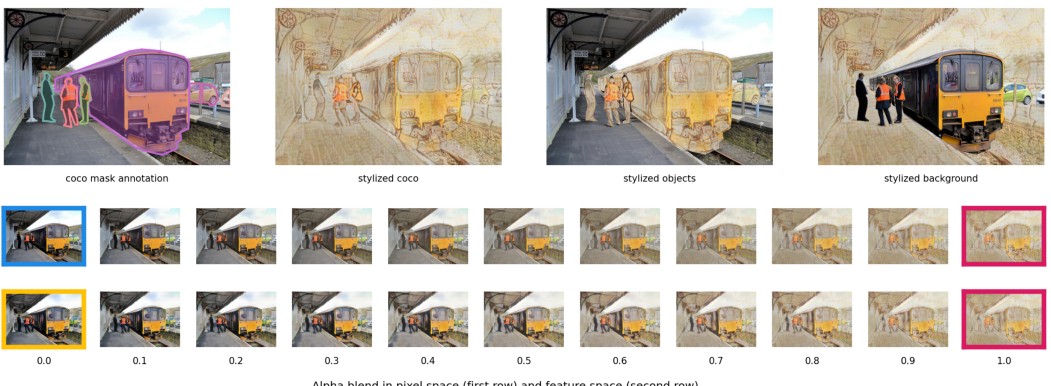

Figure 12: Comparison of Stylized COCO, Stylized Objects and Background for the last example in Figure 11. Bottom rows show the pixel and feature space blending sequences for the stylized COCO version.

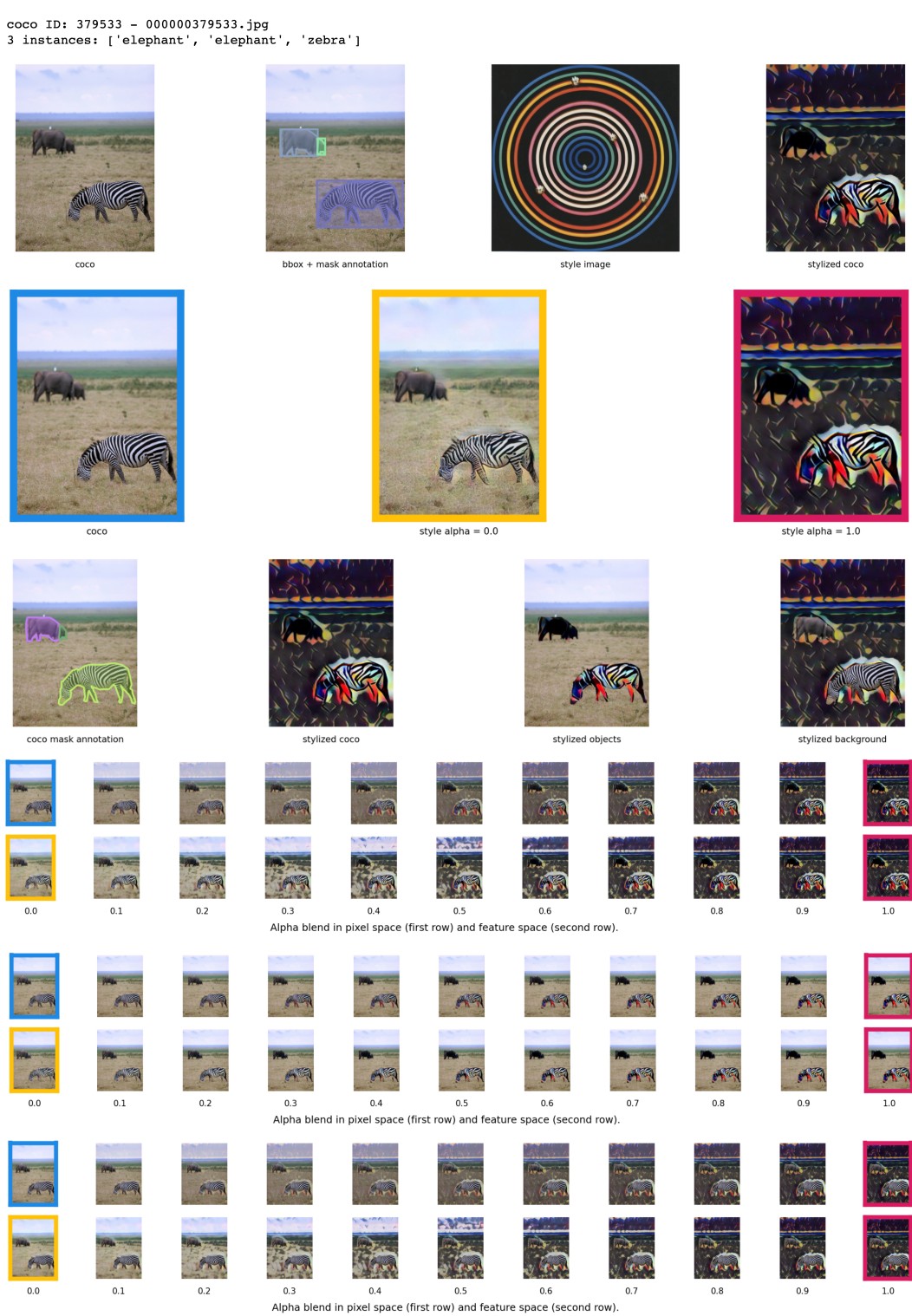

Figure 13: Full example for one image. **Top row:** Creation Process of Stylized-COCO. **Second row:** Comparison of AdaIn style strength. **Third row:** Comparison of the stylized datasets. **Last three rows:** Blending Sequences for Stylized-COCO, Stylized-Objects and Stylized-Background. We create these for every image which results in 60 modified copies of the COCO val2017 subset.

The configuration to reproduce the exact same version of stylized COCO can be found in our code release.

A NOTE ON THE ROLE OF DATA AUGMENTATION

A certainly under-explored dimension of our study is the impact of data augmentation for systematic generalization. It is known that specific augmentation techniques will improve the performance on certain problem domains. For instance, Geirhos et al. (2019) and Michaelis et al. (2019) have shown that the texture bias in some classification and detection models can be reduced when stylized images are used during training. It is not know however if the learned representations generalize in a more systematic way, e.g. improve the performance on occluded objects for instance. It is also not known if frameworks and architectures benefit equally from such augmentation or if some models can exploiting the regularized signal more efficiently. These are certainly interesting questions but would require to re-train all models that are used in our study from scratch. As this would result in a significant amount of compute, we decided against this experiment for the following reasons. First, our infrastructure is simply limited. Second, we consider the investment to be disproportional to the expected result. As can be seen from Figure 9, die difference in robustness for a ResNet-50 without and with Large Scale Jitter augmentation lies within a few percent. In contrast, frameworks and neck architectures appeared to have a much larger impact on robustness. In consequence, we opted to make these dimension a priority in our comparison and to postpone a more in depth analysis of data augmentation.

PRE-TRAINED MODEL WEIGHTS

We use code and checkpoints from the following projects. Note that unfortunately, CPMask (Fan et al., 2020) was not fully released at the time of this publication but will be included in our code release in the future. Before we test a model on Stylized COCO and its variants, we reproduced the reported score on COCO `val2017`. Models without reported metrics on `val2017` have been validated on `test-dev2017` before testing.

- Detectron 2: `https://github.com/facebookresearch/detectron2/`
- BMask R-CNN: `https://github.com/hustvl/BMaskR-CNN`
- PyContrast (SSL): `https://github.com/HobbitLong/PyContrast/`
- Swin: `https://github.com/SwinTransformer/Swin-Transformer-Object-Detection`
- YOLO: `https://github.com/AlexeyAB/darknet`
- YOLACT(++): `https://github.com/dbolya/yolact`
- DETR: `https://github.com/facebookresearch/detr`
- BCNet: `https://github.com/lkeab/BCNet`
- CPMask: `https://github.com/fanq15/FewX`
- SOTR: `https://github.com/easton-cau/SOTR`
- SOLOv2: `https://github.com/aim-uofa/AdelaiDet/tree/master/configs/SOLOv2`

ADDITIONAL RESULTS

We include additional results for comparison. In particular, we append the equivalent bounding box versions of our main paper figures. Figure 14 compares the absolute performance on COCO `val2017` between all models. The model grouping and detailed results we present in the main paper are based on the distance matrices in Figure 15. Figure 16 provides the bbox equivalent to the general overview on model robustness. We also include the absolute performance versions of those figures. Figure 17 compares the relative framework performance by bbox score. Figure 18 and Figure 19 display the comparisons of backbone and neck architectures respectively.

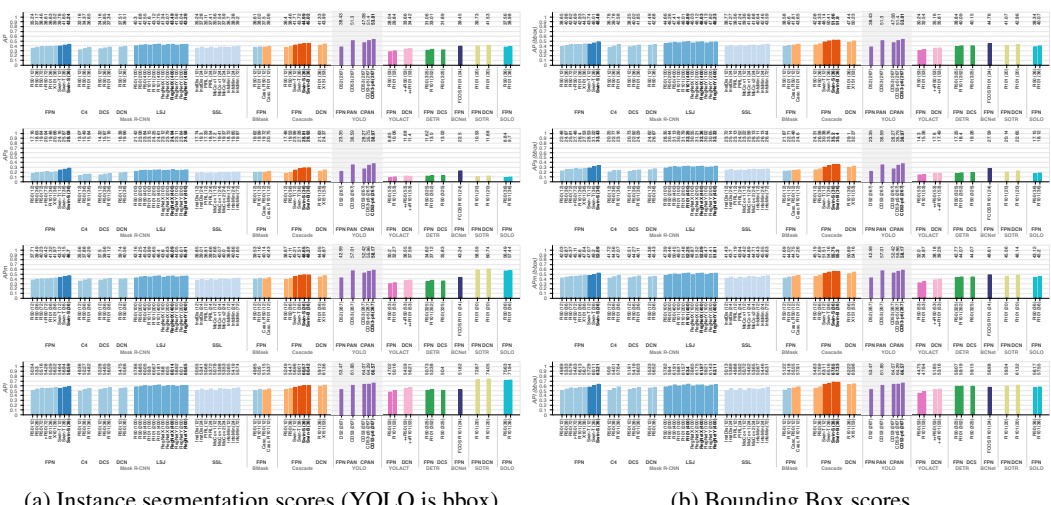

    (a) Instance segmentation scores (YOLO is bbox)          (b) Bounding Box scores

Figure 14: Comparison of absolute model performance on COCO `val2017`. Methods that do not report scores for `val2017` have been validated on `test-dev2017`. Note that SOTR and SOLO perform worse on small objects but exhibit significantly improved segmentation scores for medium and large objects

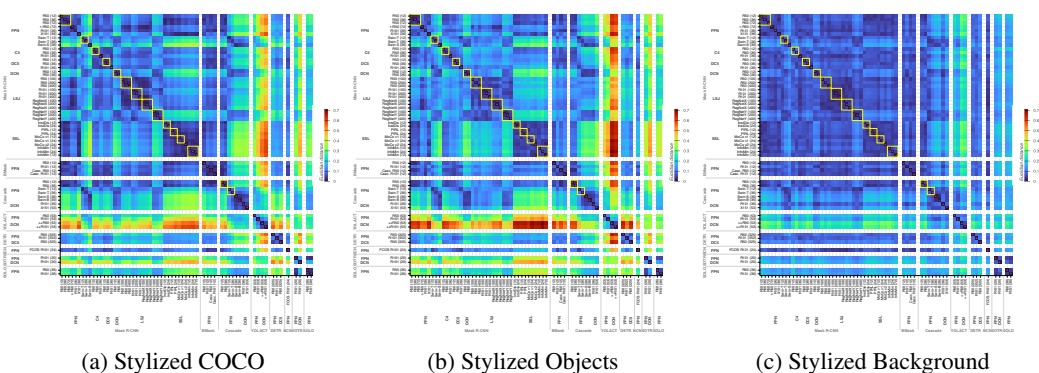

    (a) Stylized COCO          (b) Stylized Objects          (c) Stylized Background

Figure 15: Euclidian distance between the relative model performances (over the full alpha range). We average over AP, APs, APm and APl. Yellow squares group the same model with different learning schedule.

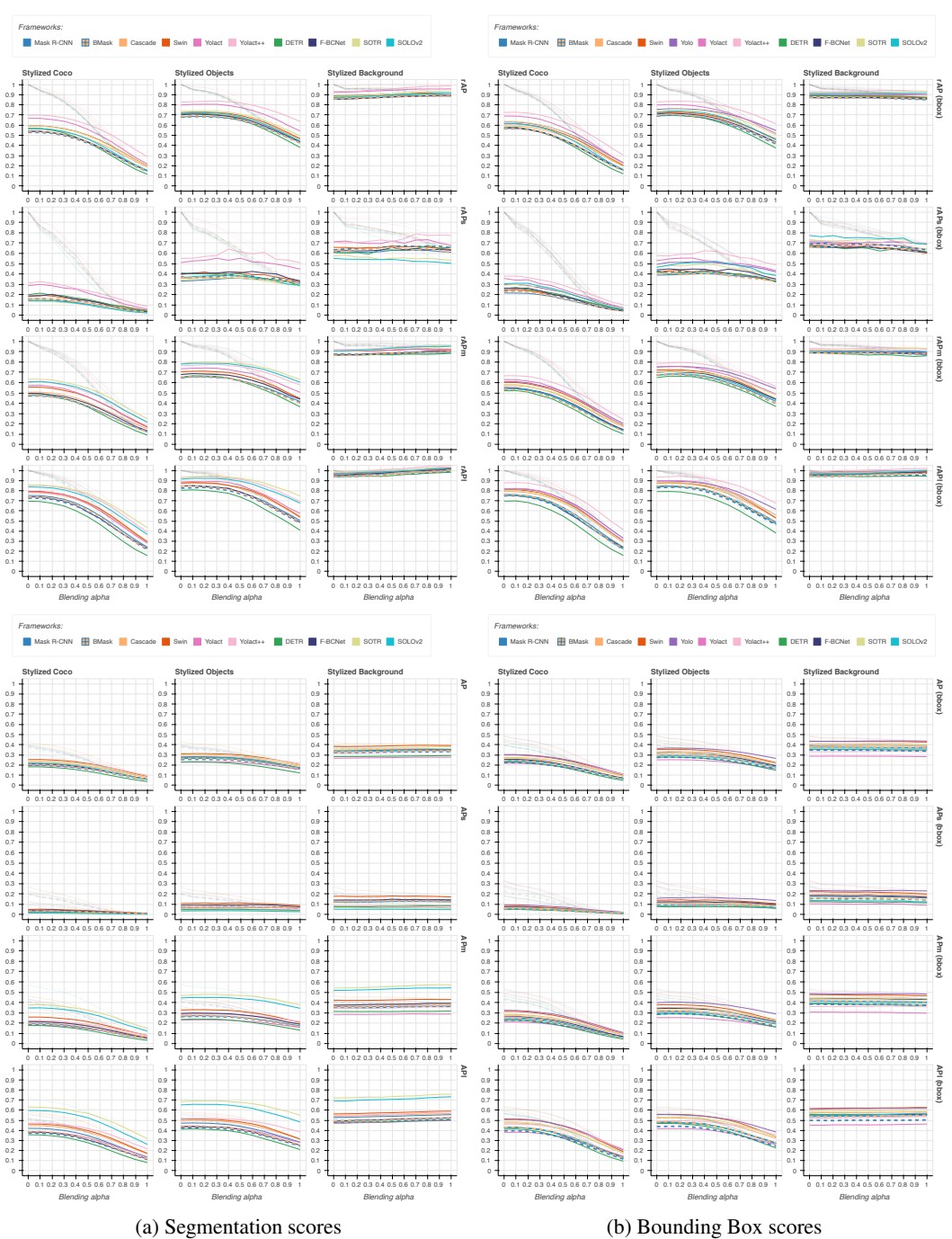

(a) Segmentation scores  (b) Bounding Box scores

Figure 16: General overview of model robustness. Top row shows relative performance as in the main paper. Bottom row displays absolute performance for comparison. Note that comparing relative or absolute performance results in a different order of frameworks due to varying base performance.

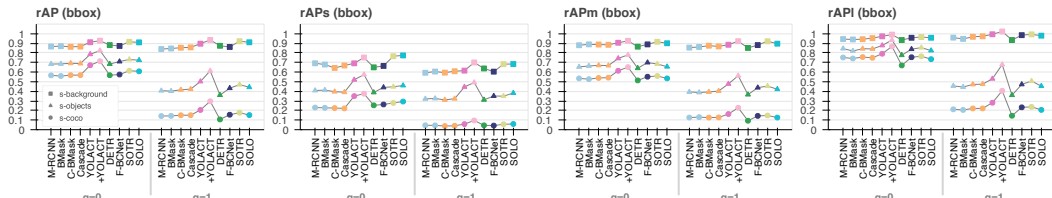

Figure 17: Object-centric robustness by framework (bbox). Note that we compromised on R-101 for SOTR.

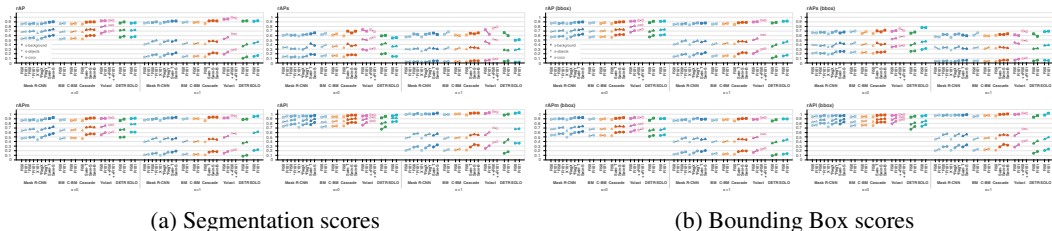

(a) Segmentation scores

(b) Bounding Box scores

Figure 18: Full comparison of object-centric robustness by backbone architecture. Models marked with * are trained with LSJ.

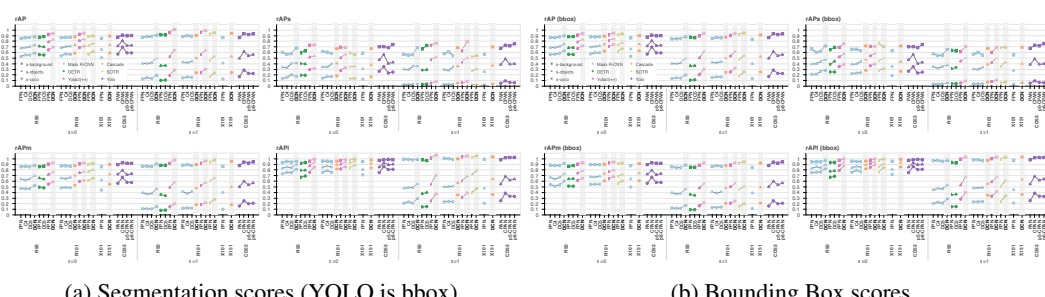

(a) Segmentation scores (YOLO is bbox)

(b) Bounding Box scores

Figure 19: Full comparison of object-centric robustness by neck architecture. Models with dynamic components are highlighted.

