# OpenReview forum: "An object-centric sensitivity analysis of deep learning based instance segmentation"
_ICLR.cc/2022/Conference — ICLR 2022 Submitted_

### Official Review · Reviewer_Hfwx · 2021-10-29

**Correctness:** 3
**Technical Novelty And Significance:** 2
**Empirical Novelty And Significance:** 2
**Recommendation:** 6
**Confidence:** 5

**Main Review:**

Strength:
1. A new object-centric evaluation dataset based on COCO val is constructed for evaluating the robustness instance segmentation models.
2. The robustness evaluation is mainly evaluated from 3 impacted dimensions, which are well categorized. Also, a huge number of evaluations and comparisons are conducted.
3. The paper is presented in a clear structure and easy to understand.

Weakness:

1. The paper evaluates the robustness of instance segmentation, mainly from the image/feature corruptions perspective. However, in real life, the robustness for instance segmentation should be more about the adaptation to novel objects [a,b] in a long-tailed distribution world or how to handle overlapping objects with similar appearance in heavy occlusions [c,d]. There should be some relations discussion with existing instance segmentation works that also focus in robust instance segmentation but from different directions.

2. Instance segmentation models can be divided into backbone, neck and functional heads. The backbone and neck have been studied, but what’s the influence of the mask head design for instance segmentation? Some popular mask head designs, besides the Mask R-CNN, include BMask R-CNN [e] and BCNet [d], which also need to be evaluated and discussed.

    [a] Learning to Segment Every Thing. CVPR, 2018.

    [b] Commonality-Parsing Network across Shape and Appearance for Partially Supervised Instance Segmentation. ECCV, 2020.

    [c] Multi-instance object segmentation with occlusion handling. CVPR, 2015.

    [d] Deep Occlusion-Aware Instance Segmentation with Overlapping BiLayers. CVPR, 2021.

    [e] Boundary-preserving mask r-cnn. ECCV, 2020.

3. The paper uses the image styling to simulate the image corruptions. How to guarantee it is realistic to real image corruptions with practical usages? For example, in Figure 1: right, a clear separation between background and foreground is observable in Stylized Object and Background but that is not common in real world, which may make the object detection and segmentation easier. I think some discussions on how to generalize the simulation to realistic world corruption cases should be added.

4. The ambiguous usage of term 'architecture' vs. 'framework'. The 'architecture' should be replaced with 'backbone' instead, as in the first column in Table 1, most variants are R50, R101, Swin-T, etc. Usually, the usages of 'architecture' and 'framework' are similar.


**Summary Of The Paper:**

The paper presents an evaluation on object-centric robustness of existing instance segmentation models, where an object-centric stylized coco val dataset is constructed. Then, a large number of experiments and evaluations are performed on this new dataset to study the impact of framework, architecture and pre-training.

**Summary Of The Review:**

My concerns are listed in the weakness section. If the authors can well address them. I would consider raising my score.

After reading the updated paper, my concern about the mask head influence is addressed. Also, I have read a new intro and abstract with better motivation connecting to the real-world instance segmentation challenges. Although image corruption using style transferring has no real-world cases, it does create novel object/background context. The experiments conducted are extensive and it provides some suggestions of instance segmentation model design for environment robustness. Thus, I decide to increase my rating to 6.

---

> ### Author Response · Authors · 2021-11-12
> **Response to 1) additional perspective 2) mask head comparison 3) style transfer corruptions 4) wording**
>
> Hello and thank you for your feedback. We are glad that you find our work clear and easy to understand. We invested a significant amount of time to make it as accessible as possible. We now address your concerns:
>
> 1) We are aware of the perspectives you mention and familiar with most work you recommend. We agreed that it would be an improvement to include it in our work and we will happily do so. The reason why we have not included it already is simply the page limit that forced us to cut some corners. As you can see from our response to reviewer 1, our focus on Gheiros et al. might have been too strong and potentially misleading. We hence plan to reduce some parts of the motivation and include the additional perspectives instead. We find that the sections 'related work' or 'discussion' are the best fit or maybe a new 'limitations' (of our work) section. What do you think? We will let you know when we uploaded an updated version of the paper.
>
> 2) Thank you for highlighting the importance of mask heads. We thought a lot about how to measure the performance of mask heads in isolation but came to the conclusion that there is no satisfying solution since the mask heads inherently depend on the backbone and neck performance. For instance, feeding ground truth proposals to the mask head would introduce an undesired bias. Now concerning your request. We were not aware of the BCNet and BMask works so thank you for pointing these out! We found model checkpoints for both and will run the experiments! The only 'fear' we have is the upcoming deadline Nov 22. If time runs out we might have to consider including it as an ablation study so we don't have to rework the main figures etc. which is surprisingly time consuming. We give our best however to include it properly and let you know about the progress beginning next week :)
>
> 3) We think that the corruption types that are introduced by the stylization process can actually not be found in real life (they are introduced by the artificial AdaIn methods CNN). Our general idea here is not simulate 'corruptions' (like noise or blur) but rather to simulate novel (unseen) appearances of familiar objects. The term 'object-centric robustness' is therefore more closely related to the often used (but ill defined) notion of 'out of distribution' generalization, i.e. does a model generalize to / is robust to a familiar object with a novel texture?. In our related work section we provide the 3DB framework (rendering engine) as an alternative to the style transfer method. Here, texture transfer can be accomplished without the introduction of corruptions/artifacts at the cost that the rendering engine introduces a certain sim-to-real gap for the model. See:
>
> - https://www.microsoft.com/en-us/research/publication/3db-a-framework-for-debugging-computer-vision-models/
> - https://github.com/3db/3db
>
> Very recently (after submitting), we found the following work as another alternative to what we aim to accomplish with the style transfer process. We may include it in our related work of datasets as well:
>
> - Adversarial Objects https://arxiv.org/abs/2111.04204v1
> - https://paperswithcode.com/dataset/nao
>
> 3.1) We like to point out, that the example in Figure 1 is chosen intentionally for its clear/sharp boundary to make the principle of masking obvious to the reader. We understand however you concern and are willing to address it in two ways. First, we will include a more extensive and representative amount of masked images in the Appendix. Second, we can provide a discussion of this problem in the Appendix and potential ways how to extend/adapt this to more real life situations. A naive approach would be to use soft edges (gaussian blur) instead of sharp edges for masking. Do you think such a discussion would sufficiently address your concern or do you request this discussion to be in the main text?
>
> 4) We understand the problem with our ambiguous usage of the terms 'architecture' vs. 'framework'. We have no problem with this request and will update our work accordingly. We think it will make things more clear.
>
> We hope our response addresses most of your concerns an we are looking forward to an ongoing and fruitful discussion. Do you think that this is going into a direction where you would consider updating your score? Thanks again for reviewing our paper.

---

> > ### Comment · Reviewer_Hfwx · 2021-11-16
> > **Thanks for the reply.**
> >
> > Hi, thanks for the detailed response. Most of improving directions are matching my review concerns.  What is the experiment progress updating for the 2nd experiments on the "the importance of mask heads"? Maybe more discussions could happen with new updated version of the paper. The robustness of instance segmentation should be better introduced in the paper by connecting the practical real-world challenges.

---

> > > ### Author Response · Authors · 2021-11-16
> > > **Preview result for BMask and BCNet 2/2**
> > >
> > > ### **Stylized Objects**
> > >
> > > | S-Objects (feature_space, segm @ 0.0) |  rAP  | rAP50 | rAP75 | rAPs  | rAPm  | rAPl  |
> > > | --- | --- | --- | --- | --- | --- | --- |
> > > | mask_rcnn_R_50_FPN_1x             | 66.9 | 70.4 | 65.1 | 33.9 | 62.9 | 83.6 |
> > > | mask_rcnn_R_101_FPN_3x            | 68.6 | 71.4 | 67.6 | 34.1 | 65.4 | 84.7 |
> > > | bmask_rcnn_r50_1x                 | 67.1 | 70.4 | 65.0 | 35.2 | 63.9 | 82.1 |
> > > | bmask_rcnn_r101_1x                | 68.8 | 71.6 | 67.5 | 39.4 | 65.9 | 83.0 |
> > > | cascade_bmask_rcnn_r50_1x         | 67.7 | 70.7 | 65.6 | 35.9 | 64.6 | 83.8 |
> > > | cascade_bmask_rcnn_r101_1x        | 69.4 | 72.1 | 68.0 | 36.9 | 66.2 | 85.1 |
> > > | fcos_imprv_R_101_FPN              | 70.1 | 72.9 | 68.2 | 40.1 | 67.8 | 84.5 |
> > >
> > > | S-Objects (feature_space, segm @ 0.5) |  rAP  | rAP50 | rAP75 | rAPs  | rAPm  | rAPl  |
> > > | --- | --- | --- | --- | --- | --- | --- |
> > > | mask_rcnn_R_50_FPN_1x             | 63.1 | 64.6 | 62.3 | 36.8 | 60.0 | 76.2 |
> > > | mask_rcnn_R_101_FPN_3x            | 64.9 | 65.9 | 63.9 | 35.9 | 61.9 | 78.4 |
> > > | bmask_rcnn_r50_1x                 | 63.3 | 64.3 | 61.8 | 35.4 | 60.1 | 75.5 |
> > > | bmask_rcnn_r101_1x                | 66.2 | 66.7 | 65.2 | 39.9 | 63.8 | 78.2 |
> > > | cascade_bmask_rcnn_r50_1x         | 64.3 | 65.5 | 62.7 | 36.2 | 61.1 | 76.6 |
> > > | cascade_bmask_rcnn_r101_1x        | 65.7 | 66.7 | 64.4 | 38.8 | 62.4 | 78.8 |
> > > | fcos_imprv_R_101_FPN              | 66.7 | 67.7 | 65.3 | 41.2 | 65.0 | 78.4 |
> > >
> > > | S-Objects (feature_space, segm @ 1.0) |  rAP  | rAP50 | rAP75 | rAPs  | rAPm  | rAPl  |
> > > | --- | --- | --- | --- | --- | --- | --- |
> > > | mask_rcnn_R_50_FPN_1x             | 40.8 | 40.9 | 40.2 | 29.2 | 39.2 | 46.9 |
> > > | mask_rcnn_R_101_FPN_3x            | 42.9 | 42.8 | 42.4 | 29.7 | 41.5 | 49.8 |
> > > | bmask_rcnn_r50_1x                 | 40.9 | 40.8 | 40.2 | 30.2 | 39.1 | 46.3 |
> > > | bmask_rcnn_r101_1x                | 43.9 | 43.6 | 43.4 | 33.8 | 41.9 | 49.1 |
> > > | cascade_bmask_rcnn_r50_1x         | 41.7 | 41.2 | 40.8 | 29.4 | 39.4 | 47.6 |
> > > | cascade_bmask_rcnn_r101_1x        | 43.4 | 43.2 | 42.7 | 32.8 | 40.8 | 49.1 |
> > > | fcos_imprv_R_101_FPN              | 43.6 | 43.4 | 43.0 | 33.4 | 43.9 | 48.6 |
> > >
> > > ### **Stylized Background**
> > >
> > > | S-Background (feature_space, segm @ 0.0) |  rAP  | rAP50 | rAP75 | rAPs  | rAPm  | rAPl  |
> > > | --- | --- | --- | --- | --- | --- | --- |
> > > | mask_rcnn_R_50_FPN_1x             | 85.4 | 89.4 | 83.4 | 62.1 | 85.1 | 94.3 |
> > > | mask_rcnn_R_101_FPN_3x            | 86.5 | 90.0 | 85.3 | 62.0 | 87.4 | 95.8 |
> > > | bmask_rcnn_r50_1x                 | 85.8 | 90.1 | 83.6 | 61.9 | 87.0 | 93.6 |
> > > | bmask_rcnn_r101_1x                | 86.8 | 90.7 | 85.3 | 64.8 | 87.9 | 94.8 |
> > > | cascade_bmask_rcnn_r50_1x         | 85.3 | 89.2 | 83.1 | 61.0 | 86.0 | 94.1 |
> > > | cascade_bmask_rcnn_r101_1x        | 86.6 | 90.4 | 84.5 | 63.5 | 87.3 | 94.3 |
> > > | fcos_imprv_R_101_FPN              | 85.6 | 90.0 | 83.5 | 61.9 | 86.6 | 94.9 |
> > >
> > > | S-Background (feature_space, segm @ 0.5) |  rAP  | rAP50 | rAP75 | rAPs  | rAPm  | rAPl  |
> > > | --- | --- | --- | --- | --- | --- | --- |
> > > | mask_rcnn_R_50_FPN_1x             | 86.6 | 88.5 | 85.2 | 64.3 | 85.3 | 95.9 |
> > > | mask_rcnn_R_101_FPN_3x            | 87.9 | 88.7 | 87.2 | 63.5 | 87.5 | 97.8 |
> > > | bmask_rcnn_r50_1x                 | 87.7 | 88.8 | 86.1 | 65.4 | 88.0 | 95.9 |
> > > | bmask_rcnn_r101_1x                | 88.5 | 89.2 | 87.5 | 67.5 | 88.7 | 96.7 |
> > > | cascade_bmask_rcnn_r50_1x         | 87.1 | 88.1 | 85.6 | 64.1 | 87.4 | 95.9 |
> > > | cascade_bmask_rcnn_r101_1x        | 88.6 | 89.5 | 87.1 | 67.3 | 89.4 | 96.4 |
> > > | fcos_imprv_R_101_FPN              | 88.0 | 89.3 | 86.4 | 65.7 | 88.6 | 98.2 |
> > >
> > > | S-Background (feature_space, segm @ 1.0) |  rAP  | rAP50 | rAP75 | rAPs  | rAPm  | rAPl  |
> > > | --- | --- | --- | --- | --- | --- | --- |
> > > | mask_rcnn_R_50_FPN_1x             | 87.3 | 84.1 | 87.9 | 59.3 | 86.9 |  99.1 |
> > > | mask_rcnn_R_101_FPN_3x            | 88.3 | 84.1 | 88.8 | 57.7 | 88.5 | 100.7 |
> > > | bmask_rcnn_r50_1x                 | 88.5 | 84.4 | 88.5 | 62.5 | 89.1 |  99.0 |
> > > | bmask_rcnn_r101_1x                | 89.6 | 84.9 | 89.6 | 68.2 | 90.3 |  99.4 |
> > > | cascade_bmask_rcnn_r50_1x         | 88.5 | 84.3 | 87.5 | 60.9 | 90.0 | 100.2 |
> > > | cascade_bmask_rcnn_r101_1x        | 89.3 | 84.7 | 88.7 | 59.8 | 90.1 | 100.6 |
> > > | fcos_imprv_R_101_FPN              | 89.5 | 85.5 | 89.3 | 63.0 | 90.7 | 101.4 |

---

> > > ### Author Response · Authors · 2021-11-16
> > > **Preview result for BMask and BCNet 1/2**
> > >
> > > Hello, we managed to test 5 additional models. Namely BMask & Cascade BMask R-50 & R-101 as well as FCOS BCNet R-R101-FPN (weights from the official repositories). We currently reworking our figures, text and the overview table accordingly. We will upload an update version of the paper later this week. This change will also include a reworked introduction, motivation and discussion as well as a short limitations chapter.
> > >
> > > We agree that we have to better introduce instance segmentation robustness as can also be seen from the other reviews. Our introduction is currently too one-sided and crucially misses a better connection to real-world challenges as you point out.
> > >
> > > As a 'preview' we append a comparison of the 5 new models with standard Mask R-CNN R-50/R-101 for the feature space sequence at alpha 0.0, 0.5 and 1.0. We can see that BMask and BCNet are slightly more robust in some cases but overall, closely follow standard Mask R-CNN results. Note that we report relative AP values in % (-> rAP).
> > >
> > > ### **Stylized COCO**
> > >
> > > | S-COCO (feature_space, segm @ 0.0) |  rAP  | rAP50 | rAP75 | rAPs  | rAPm  | rAPl  |
> > > | --- | --- | --- | --- | --- | --- | --- |
> > > | mask_rcnn_R_50_FPN_1x             | 53.0 | 60.1 | 48.9 | 14.3 | 46.7 | 73.2 |
> > > | mask_rcnn_R_101_FPN_3x            | 54.2 | 60.8 | 50.7 | 14.4 | 48.2 | 75.8 |
> > > | bmask_rcnn_r50_1x                 | 52.1 | 59.6 | 48.1 | 14.8 | 46.2 | 71.8 |
> > > | bmask_rcnn_r101_1x                | 53.6 | 60.6 | 50.3 | 16.4 | 48.0 | 73.4 |
> > > | cascade_bmask_rcnn_r50_1x         | 53.4 | 60.3 | 49.3 | 15.9 | 48.2 | 73.5 |
> > > | cascade_bmask_rcnn_r101_1x        | 54.0 | 61.5 | 49.4 | 16.1 | 47.7 | 74.1 |
> > > | fcos_imprv_R_101_FPN              | 54.1 | 61.5 | 49.6 | 18.8 | 49.6 | 73.9 |
> > >
> > > | S-COCO (feature_space, segm @ 0.5) |  rAP  | rAP50 | rAP75 | rAPs  | rAPm  | rAPl  |
> > > | --- | --- | --- | --- | --- | --- | --- |
> > > | mask_rcnn_R_50_FPN_1x             | 42.6 | 49.0 | 38.5 | 11.7 | 36.4 | 60.7 |
> > > | mask_rcnn_R_101_FPN_3x            | 43.6 | 49.9 | 40.1 | 10.5 | 37.5 | 62.2 |
> > > | bmask_rcnn_r50_1x                 | 41.4 | 48.0 | 38.0 | 12.1 | 35.9 | 57.5 |
> > > | bmask_rcnn_r101_1x                | 43.1 | 49.6 | 39.7 | 12.6 | 37.4 | 60.7 |
> > > | cascade_bmask_rcnn_r50_1x         | 42.6 | 48.8 | 39.0 | 11.9 | 37.2 | 59.6 |
> > > | cascade_bmask_rcnn_r101_1x        | 43.5 | 50.0 | 39.7 | 11.9 | 37.1 | 61.8 |
> > > | fcos_imprv_R_101_FPN              | 43.2 | 50.0 | 39.2 | 14.0 | 39.7 | 58.8 |
> > >
> > > | S-COCO (feature_space, segm @ 1.0) |  rAP  | rAP50 | rAP75 | rAPs  | rAPm  | rAPl  |
> > > | --- | --- | --- | --- | --- | --- | --- |
> > > | mask_rcnn_R_50_FPN_1x             | 13.4 | 15.9 | 11.9 | 3.3 | 11.2 | 21.4 |
> > > | mask_rcnn_R_101_FPN_3x            | 14.8 | 17.4 | 13.2 | 3.0 | 12.1 | 23.5 |
> > > | bmask_rcnn_r50_1x                 | 13.4 | 16.0 | 11.9 | 3.1 | 11.6 | 20.2 |
> > > | bmask_rcnn_r101_1x                | 15.2 | 17.9 | 13.7 | 3.8 | 13.1 | 22.8 |
> > > | cascade_bmask_rcnn_r50_1x         | 14.4 | 16.7 | 13.1 | 3.1 | 11.9 | 21.7 |
> > > | cascade_bmask_rcnn_r101_1x        | 15.2 | 17.9 | 13.8 | 3.4 | 12.4 | 23.4 |
> > > | fcos_imprv_R_101_FPN              | 14.8 | 17.4 | 13.4 | 3.3 | 13.2 | 22.8 |

---

> > > ### Author Response · Authors · 2021-11-22
> > > **We uploaded a new version of the paper!**
> > >
> > > Hello, we uploaded a new version of the paper. You can find the list of changes here (general comment on top):
> > >
> > > https://openreview.net/forum?id=C5Q04gnc4f&noteId=R9ist2EwoJ3

---

> > > > ### Comment · Reviewer_Hfwx · 2021-11-24
> > > > **Rely to the Updated Paper**
> > > >
> > > > Hi, thanks for the updated papers. After reading the updated paper, my concern about the mask head influence is addressed. Also, I have read a new intro and abstract with better motivation connecting to the real-world instance segmentation challenges. Although image corruption using style transferring has no real-world cases, it does create novel object/background context. The experiments conducted are extensive and it provides some suggestions of instance segmentation model design for environment robustness. Thus, I decide to increase my rating to 6.
> > > >
> > > > As another reviewer suggested, what's the influence if the noise/jpeg artifacts/motion blur etc applied on the images instead of the style transferring by AdaIN? Could we expect similar experiment results and conclusion in this case?

---

> > > > > ### Author Response · Authors · 2021-11-26
> > > > > **Thank you for reviewing our paper**
> > > > >
> > > > > Thank you for reviewing our paper! We are happy to hear that we have well addressed your concerns and we appreciate the updated score. The paper has been certainly improved with your feedback.
> > > > >
> > > > > We prepared a little ablation study regarding the impact of different corruption types here:
> > > > >
> > > > > https://openreview.net/forum?id=C5Q04gnc4f&noteId=Lcg0ptna8LN
> > > > >
> > > > > Feel free to chime in if you have additional comments or questions on this topic.

---

### Official Review · Reviewer_GTkh · 2021-11-01

**Correctness:** 1
**Technical Novelty And Significance:** 1
**Empirical Novelty And Significance:** 1
**Recommendation:** 3
**Confidence:** 5

**Main Review:**

Strengths:
The authors compare a variety of instance segmentation techniques along different axes like frameworks, architecture and pre-training

Concerns:
My concern with the approach is that because it uses ground truth masks to generate different versions of datasets, it creates a bias where the features are corrupted, and this affects the performance of detectors. The RGB features inside the object masks are not changed in the stylized object dataset (apart from the minor corruption of features from the background), so it is expected that detectors would perform best in this case. When we start changing the RGB features of the objects but keep the background as is, the performance degrades, although not entirely because object boundaries would still be visible to some extent. Finally, when the entire image is stylized, the boundaries get mixed up and there is very little signal in the image (as can be seen Figure 2), so the detector does not work in this case. Overall, it remains unclear to me if the method is adding any insights compared to what is not already expected. I expect similar problems if any other transformations (like gaussian noise, salt/pepper noise/blur etc.) are applied to COCO images on object masks, background and entire image. Also, its unclear if there is something special about using stylized transformations to make claims that object detectors are learning shape/contour specific features - the conclusion which I am taking here is “the more noise we add, the worse object detectors get”, more specifically, noise everywhere > noise on object > noise on background.

Other Comments:
Figure 6 is very dense and hard to read. What are the light/dark lines in stylized coco/ stylized objects?

The authors should include 20-30 examples of stylized images in the paper or the supplementary material. This will give better insights to the reader about how the dataset looks in different cases.


**Summary Of The Paper:**

In this paper, the authors study whether deep learning based techniques for instance segmentation are robust to changes in object texture or contour. This is inspired from a previous work in ICLR 2019 where the authors perform a similar study for image classification and show that existing CNN models learn classes based on texture and do not take the overall shape into account.

The authors take the task of instance segmentation where it is necessary for the model to learn the overall shape and measure the impact of texture changes. They use a stylized version of coco but also create two different types, one stylizing only the objects and the other stylizing only the background. Moreover, for each dataset type they create 10 variants by varying the blending weight. The evaluation is done on four dataset types, default coco val, stylized coco (everything stylized), stylized objects and stylized images. Except the default coco, every dataset type has 20 variants (different levels of blending plus pixel space vs feature space blending) which leads to a total of 61 datasets.
Following this, the authors perform an evaluation of various instance segmentation techniques. They take three paradigms - architectures (backbone / neck), frameworks and pre-training approaches and study the robustness of each to changes in texture.


**Summary Of The Review:**

There are gaps in the experimental design based on which we are concluding what features object detectors learn.

---

> ### Author Response · Authors · 2021-11-12
> **Response to 4) results can be expected 5) novelty and significance**
>
> 4) We address your final conclusion “the more noise we add, the worse object detectors get” as a separate point. We agree that this can be expected. The idea behind our paper is not to simply show that detectors degrade with increasing corruption (which can be expected). Our main contribution is the extensive and rigorous comparison between frameworks, architectures and pre-training schemes (i.e. do different frameworks or backbones degrade differently?). The result is a solid baseline that unveils a non trivial ordering of models regarding object-centric robustness/generalization. For instance, why is YOLACT almost consistently more robust than all other frameworks? Why are dynamic convolutions more robust than fixed kernels? etc. It is not clear to us how such findings could be expected in advance. We consider these findings valuable for the community and hope that they motive more detailed research in this direction. We understand that we could make our intention more explicit to the reader if you consider that an improvement? If you disagree with our reasoning, can you provide more details or maybe literature which lets the mentioned findings be trivially expectable?
>
> Finally, the following remarks are the same for Reviewer 2 and 3 who raised similar concerns (and similar rating). It might be worthwhile reading the others review and our individual response to it. By now we hope to have addressed your concerns well enough to start a fruitful discussion. In case you find our arguments somewhat convincing, do you think that this is heading in a direction where, at some point, you would consider updating your score?
>
> 5) We were a little surprised by your ratings of 1 (The contributions are neither significant nor novel). To the best of our knowledge, there exists no prior work that investigates the object-centric (causally plausible) robustness of instance segmentation models at this scale (in fact, not at all). Regarding significance of results we like to point out that the current SOTA methods (1,2) on COCO val2017 improve +1% and +6% (49.5 and 51.9 mAP, single scale testing) over the previous SOTA method (3) (48.9 mAP). In comparison, we find that models vary significantly more and in non trivial ways regarding robustness. For instance, YOLACT is almost consistently more robust than other methods with a difference of ~10-30% relative mAP. A similar argument can be made for SOTR and SOLOv2 etc. Instead of simply chasing the next SOTA score, we hope that our results contribute to and encourage more research in the direction of robustness and non trivial generalization.
>
> (1) ICCV 2021: Swin Transformer: Hierarchical Vision Transformer Using Shifted Windows https://openaccess.thecvf.com/content/ICCV2021/html/Liu_Swin_Transformer_Hierarchical_Vision_Transformer_Using_Shifted_Windows_ICCV_2021_paper.html
>
> (2) ICCV 2021: End-to-End Semi-Supervised Object Detection With Soft Teacher https://openaccess.thecvf.com/content/ICCV2021/html/Xu_End-to-End_Semi-Supervised_Object_Detection_With_Soft_Teacher_ICCV_2021_paper.html
>
> (3) CVPR 2021: Simple Copy-Paste Is a Strong Data Augmentation Method for Instance Segmentation https://openaccess.thecvf.com/content/CVPR2021/html/Ghiasi_Simple_Copy-Paste_Is_a_Strong_Data_Augmentation_Method_for_Instance_CVPR_2021_paper.html

---

> ### Author Response · Authors · 2021-11-12
> **Response to 1) unclear figure 2) adding example images 3) ground truth masks and the utility of style transfer as a method**
>
> Hello and thank you for your feedback. We address your concerns starting with the easy ones first:
>
> 1) Figure 6 compares the average relative performance per model group on the feature space blending sequence (dark) and the pixel space control group sequence (light). In addition, we plot the raw values for every model in the background (very light) to give an impression of variance. We also experimented with plotting the standard deviation per group as a tube but found it to be more cluttered and less clear. We understand that it still might be confusing and are willing to remove the very light model lines from the background. Do you think that would be an improvement to make the figure more clean and accessible?
>
> 2) Adding more examples of stylized images to the supplementary material is a very good idea and we are happy to incorporate these in an update version of our work. We will let you know when we uploaded a new version in the coming days.
>
> 3) We understand that your main concern is the use of ground truth mask information and the use of style transfer as a method to understand what object detectors learn. We can provide the following thought process to start a discussion regarding your concerns:
>
> - The use of ground truth masks to limit stylization: As pointed out by you, one can expect a detector to degrade with increasing image corruptions. For the mentioned corruption types gaussian noise, salt/pepper and noise/blur etc. it is plausible to assume a global causal model (affecting the entire image) since these corruptions are likely the result of a globally applicable process (camera lenses, image sensor, etc.). In our work we instead assume an object-centric causal model, i.e. we want to understand how models generalize to novel appearances of 'familiar' instances. Without the causally plausible control groups (stylized objects/background), we can not be certain that the change in performance is actually induced by the altered instance or a spurious correlation in the context. We find for instance that detectors are significantly influenced by context information for small objects. A sub-finding that could be 'assumed' in advance but not be proven using only a globally stylized dataset (as done in many related works). We understand that every dataset in isolation is biased which is why we use all 4 versions in comparison to draw our conclusions.
>
> - Style Transfer as a method to understand what detectors learn: In contrast to blur or noise, style transfer methods 'replace' the original texture of an image/object with a novel and unseen texture. At the same time, the general shape and contour of objects is mostly preserved. 'Mostly', because this process is not without a certain degree of general feature corruption. Note that our two blending sequences enable (for the first time) that we can effectively distinguish the impact of these style artifacts and the actual transfer of unseen texture. While the former could be seen as a form of signal noise, the latter provides an artificial test case for novel but plausible object appearances. In our work we show that the signal noise dominates the performance on small objects and texture transfer dominates for medium and large objects. Without the blending sequences, we agree that it would not be possible to draw such conclusions. Since you are sceptical about the utility of style transfer, we like to point out some related work that use style transfer in a similar context:
>
> - Shape or Texture: Understanding Discriminative Features in CNNs (ICLR 2021) https://openreview.net/forum?id=NcFEZOi-rLa
> - Inverting and Understanding Object Detectors https://arxiv.org/abs/2106.13933v1
> - Are Transformers More Robust Than CNNs? https://arxiv.org/abs/2111.05464

---

> > ### Comment · Reviewer_GTkh · 2021-11-22
> > **response**
> >
> > Thanks for the comments. I have read the response.
> >
> > object-centric causal model, i.e. we want to understand how models generalize to novel appearances of 'familiar' instances?
> >
> > I don't understand what it means by "novel appearances of 'familiar' instances".
> >
> >
> > We find for instance that detectors are significantly influenced by context information for small objects.
> >
> > For small objects, the convolutional features do not have enough resolution in the feature space, so information from background affects features of objects more.
> >
> > Style Transfer as a method to understand what detectors learn: In contrast to blur or noise, style transfer methods 'replace' the original texture of an image/object with a novel and unseen texture.
> >
> > If the same experiments are performed with blur or noise, in my opinion we should get similar results as presented in this paper. If not, we should show experimentally that adding noise/blur everywhere does not hurt the performance of detectors when compared to just adding it in the background and such a thing only happens when performing style transfer. Blur/Noise also preserve the structure/shape of objects.
> >
> >
> > To the best of our knowledge, there exists no prior work that investigates the object-centric (causally plausible) robustness of instance segmentation models at this scale (in fact, not at all).
> >
> > I don't disagree that there does not exist prior work on this and its a good problem to address. However, my concern is that the solution proposed in this paper to address this problem is not technically sound and therefore the inferences made about the performance of different object detectors may not be correct.
> >
> >
> > For instance, why is YOLACT almost consistently more robust than all other frameworks? Why are dynamic convolutions more robust than fixed kernels? etc. It is not clear to us how such findings could be expected in advance.
> >
> > I don't think the paper found the root cause for why YOLACT/dynamic convolutions did better. A way to measure this would be to find the reason why YOLACT did better, and then applied that thing which 'YOLACT' did better to detector x/y/z and show that now detector x/y/z does better on those factors. Also, one should measure how much the baseline improves if we apply that feature in general (like dynamic convs) and how much it improves on some transformed image.
> >
> > Also, all these orderings are on arbitrary transformations which do not happen in real images (style transformation on whole image, object, background) and we find that detector x/y/z does better than detector a/b/c in this setting. It would be useful to know which detectors are more robust to texture vs object shape (which this paper tried to address), but I am not convinced if the technique used to bring out which factors affect shape/texture are correct.

---

> > > ### Author Response · Authors · 2021-11-22
> > > **Response & New version of the paper**
> > >
> > > Thank you for the response. Please note that we uploaded a new version of the paper. You can find the list of changes here (general comment on top):
> > >
> > > https://openreview.net/forum?id=C5Q04gnc4f&noteId=R9ist2EwoJ3
> > >
> > >
> > > 1. In this update we clarify what we mean by "novel appearances of 'familiar' instances". More precisely, "we investigate the impact of increasingly novel object texture while controlling for the effect of corrupted color, shape and contour features". We do this on 'familiar' objects in the form of learned COCO classes, i.e. by comparing to the performance on the uncorrupted test set. A related but slightly different approach would be to consider robustness as the adaptation to 'novel classes' as done by Hu et al., 2018b.
> > >
> > >
> > > 2. We also clarify our intention which is to find and unveil new and interesting research directions. With the evidence we present, we can certainly not find the root cause of why YOLACT, dynamic convolutions as well as SOTR and SOLOv2 are significantly more robust. We agree that these are interesting new directions but we can only do one step at a time and provide hypothesis in our discussion. Please note that these new directions are the result of our robustness analysis and can not be derived from standard benchmark scores.
> > >
> > >
> > > 3. Regarding the improvement of baseline performance vs. robustness we can give the following example. From Figure 5. we can tell that a ResNet-50 Mask-RCNN model with FPN neck and deformable convolutions (36 epochs), improves 3.6% over the FPN baseline without deformable convolutions (38.51 > 37.17 AP) on val2017. On the transformed images, the DCN model performs about 11,5% better (28.0 > 25.1 AP) than the FPN baseline at alpha=0 (subtle shape and texture artifacts) and about 24,6% better (18.7 > 15.0 AP) alpha=1 (out-of-distribution texture). We conclude that deformable convolutions improve robustness to feature corruptions but even more to novel texture.
> > >
> > >
> > > 4. Concerning small objects we agree and hypothesize (in our paper) "that the difference between shape and texture simply collapses for (very) small objects". Consequently we do not use the results on small objects in our conclusions about systematic generalization.
> > >
> > >
> > > 5. We understand your concern that our simulation is not technically sound and therefore, the inferences may not be correct. As a defense, we like to emphasize our in depth analysis of the stylized dataset in figure 4 where we calculate the structural similarity index and the wasserstein distance between RGB histograms. In combination with the qualitative examples, we can now tell the differentiate between shape artifacts (at the beginning of the blending sequence -> corrupted edges, same color) and actual texture transfer (towards the end of the blending sequence -> similarly corrupted edges but with different color). By inspecting the results on Stylized Objects rAPm and rAPl in figure 6 (middle column, row 3 and 4), we observe that models vary 'only' about 12-15% in relative performance at the beginning of the sequence (shape artifacts) but much more towards the end of the sequence ~25-35% (texture transfer). In summary with our analysis of the data, we conclude that the former score range captures the robustness to feature corruptions while the latter describes the robustness to out-of-distribution texture.
> > >
> > > We have to admit, that we don't fully understand how a similar experiment with noise/blur would provide more insights. How are these corruption types comparable to an artificial style transfer process? We understand however what you propose and we will run the experiment nevertheless. Due to the approaching deadline, the results from this ablation study could unfortunately only be included in our later code release and not in the rebuttal though.
> > >
> > > Thanks again for the response. We feel like we already learned a lot from answering to your sharp critique. You address critical points that should be discussed rigorously and we like to emphasize that we genuinely appreciate the time you invested so far.

---

> > > > ### Comment · Reviewer_GTkh · 2021-11-22
> > > > **reply**
> > > >
> > > > We have to admit, that we don't fully understand how a similar experiment with noise/blur would provide more insights. How are these corruption types comparable to an artificial style transfer process?
> > > >
> > > > The point I was trying to make here is that instead of doing style transfer, if we add noise/jpeg artifacts/motion blur etc. I would expect similar results. One of the main claims is that if perform style transfer on background only, detectors typically perform better than if we do style transfer on objects and doing it on the entire image is worse than each of these and if we read the paper, it seems like this finding is a major contribution of this paper. However, if we replace the word style transfer with “noise”, it is something we would already expect.
> > > >
> > > > If we just do the full image version, that’s fine (probably we get to know which detectors do better when we apply style transforms on an image), but I am not sure if only object or only background versions tell us something new, because we should expect similar directional movements if we did those experiments with arbitrary local transformations.

---

> > > > > ### Author Response · Authors · 2021-11-26
> > > > > **We prepared an ablation study with salt and pepper noise**
> > > > >
> > > > > Thank you for the extended explanation! It certainly helped us and we prepared an anonymized ablation study in response to your concerns.
> > > > >
> > > > > **TLDR**: We can show that corruptions from style transfer are not comparable to salt-and-pepper noise and that models are significantly more robust to masked style transfer than to masked noise corruptions (despite having a similar poor base performance on the fully corrupted datasets).
> > > > >
> > > > > Before you click the link and skip to the results, allow us to resolve what we believe is a little misunderstanding and to provide some concluding remarks on our method regarding style transfer corruptions.
> > > > >
> > > > >
> > > > > - The potential misunderstanding: We do not claim (at any point in the paper) that it is unexpected (or a key result) that models perform better on masked corruptions, i.e. that "masked to background" > "masked to objects" > "fully corrupted image". The reason why we added an object-centric version is the vanishing objects problem as explained in detail in section 2.1 of our main paper. The problem with fully stylized images is that they introduce an unreasonable camouflage setting (style texture everywhere) and we believe that object causal masking is simply the correct way to control and test this. The "masked to background" version is a 'cheap' byproduct but should be reported nonetheless for completeness. A similar argument can be made for the blending sequences. Note that this approach can be understood as a critique of previous methods who somewhat 'hide' the shortcomings of style transfer. In contrast, we report and control them in the most transparent way possible.
> > > > >
> > > > >
> > > > > - Our main contribution is the extensive robustness comparison of now 68 popular instance segmentation frameworks and architectures. Our objective is to identify promising trends that enable more in depth research in the future.
> > > > >
> > > > >
> > > > > - From a statistical point of view, the key difference between arbitrary local corruptions (such as noise) and style transfer is that the former follows an independence assumption (i.e. no influence from the base signal) whereas the latter produces corruptions that are highly correlated with the shape features of the content image and the texture features of a style image. Style transfer is however by no means perfect and has to be considered a noisy simulation, hence our proposed sensitivity analysis. It is however the only method that allows us to change the texture of objects in natural images within a reasonable time frame (i.e. without manually photoshopping thousands of instances).
> > > > >
> > > > >
> > > > > - Since you mention motion blur, we like to point out that this is again a different type of corruption. It is either caused by moving the camera with long exposure times or by fast moving objects. The former will lead to a fully blurred image while the latter will result in blurred objects with corrupted shape. In the literature, this problem is typically studied in sequential video data.
> > > > >
> > > > >
> > > > > - Another corruption type we could consider is occlusion. It can be expected that increasing occlusion (with random black boxes for instance) will result in a decrease of performance. However, we are sure that you would agree that this setting would test a different kind of robustness and is not comparable to gaussian noise for instance.
> > > > >
> > > > >
> > > > > The point we want to make is that different kind of corruptions will inevitably lead to poor performance at some point (we can expect similar directional movements). The specific corruption types are however not comparable and models will exhibit different performance and different robustness against different types of corruption. We can show this in a little ablation study with Salt and Pepper noise (please download if vector graphics are not displayed correctly in the browser preview):
> > > > >
> > > > > https://drive.google.com/file/d/1tVQPWa-MIlubOzyOAz6auze-hzXy33Cl/view?usp=sharing
> > > > >
> > > > > Our key insights are:
> > > > >
> > > > > 1. Style Transfer and Salt and Pepper Noise can be similar devastating but lead to different results in terms of robustness, e.g. different ranking of models.
> > > > >
> > > > > 2. Models can exploit masked style transfer much better than masked noise corruptions. While the directional movements are similar, this adds additional insights and confirms our object-centric approach.
> > > > >
> > > > > We obviously hope that you consider our updated main paper, together with the ablation study and our explanations a satisfactory response to your concerns. We are not sure about the exact procedure but if possible, we can offer to include a full ablation study in a potential camera ready version of the paper. Independent of your final decision however, we like to thank you for the constructive discussion. We feel like we learned a lot from this exchange and we consider this a win, independent of the final outcome.

---

### Official Review · Reviewer_uJDe · 2021-11-02

**Correctness:** 4
**Technical Novelty And Significance:** 1
**Empirical Novelty And Significance:** 1
**Recommendation:** 3
**Confidence:** 2

**Main Review:**

Strengths:
+ This paper conducts a large number of experiments and presents a comprehensive instance segmentation literature review.
+ The figures are very clear.

Weaknesses:
- I don't see any technical contribution for the method presented in the paper. It simply alters the dataset with existing style transfer method and perform other literature methods on the altered dataset.
- The main experiment that this paper presents is to use the style tranfered dataset. However, I don't see any user cases for this setting. Besides, the dataset is modified by using groundtruth instance labels, which makes the experiment make less sense.

**Summary Of The Paper:**

This paper presents a new setting for instance segmentation and compares a set of literature instance segmentation methods on this new setting. The new settings are: 1) use AdaIN (a style transfer method) to create stylized image, 2) only perform AdaIn on objects (use COCO's ground truth instance mask), 3) only perform AdaIN on background. This paper presents the instance segmentation AP numbers for many state-of-the-art methods (e.g. mask rcnn, swimtransformer, DETR) on the three settings with different ratio of the style transfer.

**Summary Of The Review:**

I prefer to reject this paper. My main concern of this paper is the lack of contribution to the community. To me, this paper mainly conducted a set of experiments of existing methods on a new setting. And the new data setting doesn't make a lot of sense since the groundtruth is encoded in the image and there's no user cases for such setting.

---

> ### Author Response · Authors · 2021-11-12
> **Response to 3) encoding of ground truth information 4) novelty and significance**
>
> 3) Regarding the encoding of ground truth information we can provide the following thought process. All models have seen the ground truth labels from the training data and are then tested on modified version of COCO val2017. It is true that with increasing style strength, the test data somewhat encodes ground truth information in the stylized objects and background datasets. However, this is still ground truth information that has never been seen during training and more importantly, this information is encoded in the form of unseen textures from the style transfer process. The whole setting can be understood as a negative test for the model. If the model has learned to abstract away texture details it could potentially exploit the encoded ground truth contour information and score better. However, we don't see such behavior with increasing style strength (i.e. with increasing ground truth encoding). Instead, all frameworks are severely affected by increasing style strength. Importantly, some frameworks have shown to be significantly more robust than others (e.g. YOLACT, SOTR, SOLOv2). In case we missed something or you in case you disagree with our reasoning, could you provide more details on why exactly you think that this setting does not make a lot of sense?
>
> Finally, the following remarks are the same for Reviewer 2 and 3 who raised similar concerns (and similar rating). It might be worthwhile reading the others review and our individual response to it. By now we hope to have addressed your concerns well enough to start a fruitful discussion. In case you find our arguments somewhat convincing, do you think that this is heading in a direction where, at some point, you would consider updating your score?
>
> 4) We were a little surprised by your ratings of 1 (The contributions are neither significant nor novel). To the best of our knowledge, there exists no prior work that investigates the object-centric (causally plausible) robustness of instance segmentation models at this scale (in fact, not at all). Regarding significance of results we like to point out that the current SOTA methods (1,2) on COCO val2017 improve +1% and +6% (49.5 and 51.9 mAP, single scale testing) over the previous SOTA method (3) (48.9 mAP). In comparison, we find that models vary significantly more and in non trivial ways regarding robustness. For instance, YOLACT is almost consistently more robust than other methods with a difference of ~10-30% relative mAP. A similar argument can be made for SOTR and SOLOv2 etc. Instead of simply chasing the next SOTA score, we hope that our results contribute to and encourage more research in the direction of robustness and non trivial generalization.
>
> (1) ICCV 2021: Swin Transformer: Hierarchical Vision Transformer Using Shifted Windows https://openaccess.thecvf.com/content/ICCV2021/html/Liu_Swin_Transformer_Hierarchical_Vision_Transformer_Using_Shifted_Windows_ICCV_2021_paper.html
>
> (2) ICCV 2021: End-to-End Semi-Supervised Object Detection With Soft Teacher https://openaccess.thecvf.com/content/ICCV2021/html/Xu_End-to-End_Semi-Supervised_Object_Detection_With_Soft_Teacher_ICCV_2021_paper.html
>
> (3) CVPR 2021: Simple Copy-Paste Is a Strong Data Augmentation Method for Instance Segmentation https://openaccess.thecvf.com/content/CVPR2021/html/Ghiasi_Simple_Copy-Paste_Is_a_Strong_Data_Augmentation_Method_for_Instance_CVPR_2021_paper.html

---

> ### Author Response · Authors · 2021-11-12
> **Response to 1) technical contribution 2) lack of use case**
>
> Hello and thank you for your feedback. We are happy to hear that you find our figures very clear. We invested a non trivial amount of time in testing different ways how to present our results as clearly as possible. We now address your concerns:
>
> 1) We understand that the techniques we use may appear 'simple' compared to other research and we'd like to elaborate our thinking here in more detail. The problem we face is the inherent complexity of instance segmentation frameworks. For instance, evaluating such models is already significantly more complex than evaluating the classification setting (as done in most related work). Instead of introducing even more complexity on top of this setting we opted instead for an intentionally 'simple' but well established standard approach that is easy to follow. The idea is to provide a broad and solid baseline in order to guide more 'complex' and in depth research in the future (e.g. why is YOLACT almost consistently more robust than all other frameworks? Why are dynamic convolutions more robust than fixed kernels? etc.). To the best of our knowledge, the community is lacking such a broad and causally well motivated benchmark. As we show in our work, it effectively uncovers new and interesting research directions. Would you consider it an improvement if we present this kind of reasoning more prominent and clearly at the beginning of our work?
>
> 2) We understand that you are concerned by the lack of use cases for a stylized setting. We agree that there is no obvious use case in real life as opposed to classical robustness research that is focused on gaussian blur, noise etc. (i.e. these corruptions could be induced by the signal processing of a camera sensor for instance). Our motivation to understand robustness against a causally justified stylization process is grounded in human vision (hence the strong motivation by Gheiros et al.). Humans are able to build visual abstractions that allow them to robustly adapt and generalize to uncommon settings. The stylization process resembles such a fictional test setting as it alters the texture appearance of objects while keeping the general shape (mostly) intact. As shown in our work, we can effectively control for and distinguish between the impact of stylization artifacts (could be seen as signal processing artifact) and the actual style transfer (novel and unseen texture). In the literature, the latter idea is often described by the (not very well defined) concept of "out of distribution" generalization. Note that the distinction between style artifacts and actual 'out of distribution' texture is only possible due to the blending sequences we introduce in our work. To the best of our knowledge, we are the first to properly control for this and report results on the corrected problem setting (in particular for instance segmentation). The 'use case' we aim for is to understand if and how models generalize in these settings. Other recent works that use style transfer techniques to understand model failures, robustness and generalization are:
>
> - Shape or Texture: Understanding Discriminative Features in CNNs (ICLR 2021) https://openreview.net/forum?id=NcFEZOi-rLa
> - Inverting and Understanding Object Detectors https://arxiv.org/abs/2106.13933v1
> - Are Transformers More Robust Than CNNs? https://arxiv.org/abs/2111.05464
>
> Does this address your concern about the lack of use cases? i.e. do you think that understanding object-centric generalization behavior is (or could be) a valid use case in itself?

---

> ### Author Response · Authors · 2021-11-22
> **We uploaded a new version of the paper!**
>
> Hello, we uploaded a new version of the paper. You can find the list of changes here (general comment on top):
>
> https://openreview.net/forum?id=C5Q04gnc4f&noteId=R9ist2EwoJ3

---

### Official Review · Reviewer_kn4q · 2021-11-02

**Correctness:** 3
**Technical Novelty And Significance:** 2
**Empirical Novelty And Significance:** 3
**Recommendation:** 6
**Confidence:** 3

**Main Review:**

Strengths:

The paper is a good read, and easy to follow. The paper also covered many of the popular instance segmentation frameworks and architectures. This is a good reference for researchers who address the object-centric robustness.

My main concerns are the following:

1- The paper is heavily influenced by the work in Geirhos et al. to the point that makes this paper a small incremental work.
2- The authors missed providing some explanation for the results sometimes. (e.g. yolact is consistently more robust against stylization compared to other architectures, yolov4-csp is more robust compared to rest of yolo models, and Swin being the most robust when combined with maskrcnn framework).
3- Since the paper is heavily relying on the stylized images for evaluation, and since the models are trained with different data augmentation strategies which might have influenced the robustness against stylization, I am surprised that the authors didn't study how data augmentation would influence the robustness metric. For instance, would we consider yolact more robust because of some data-augs used during training, or is it because of something else?




**Summary Of The Paper:**

The authors studied the "object-centric" robustness of various instance segmentation models. They used stylized objects/background to evaluate the performance which is adopted from a work by Geirhos et al. [1]. The authors performed large number of experiments to evaluate different aspects of instance segmentation models including framework, architecture, and pre-training.

**Summary Of The Review:**

The paper is a good study of existing deep-learning instance segmentation models, and frameworks. I enjoyed reading the paper. The authors put a lot of effort on performing large number of experiments to benchmark various models, however they put little effort on deeply studying and analyzing the results and they left it to the reader to interpret the results.

---

> ### Author Response · Authors · 2021-11-12
> **Response to 1) incremental work 2) missing explanations 3) data augmentation**
>
> Hello and thank you for your feedback. We are glad you enjoyed the read as we put a lot of effort into making the work as accessible as possible. We hope it did not end up appearing too trivial by 'hiding' most of the inherent complexity of instance segmentation frameworks. We now address your concerns:
>
> 1) We understand that at a first glance, our work can appear as an increment of Geirhos et al. We like to point out however that our approach is only *motivated* by their work and that our method and experimental setup has in fact, almost zero overlap with their work. The main contribution of Gheiros et al. is a comparison of human performance and classification models on a set of different datasets. One of these is a stylized cue conflict dataset (containing 1280 cue conflict images) to measure the texture shape bias in humans an machines. This is a different dataset than Stylized-ImageNet which was 'only' used as a data augmentation technique to reduce texture bias (measured on the cue conflict dataset). Our object-centric sensitivity analysis is a completely different approach that enables a causally plausible evaluation of robustness and abstraction (which we did not find to happen). Would you agree with our thinking here? We like to add that our emphasis on Geirhos et al. might be too strong and we are willing to make the differences more explicit for the reader if you think that would be an improvement.
>
> 2) Thank you for pointing that out! We will add the missing explanations. Note that we have been intentionally very cautious with the interpretation of results. The frameworks we compare are very complex and partially implement completely different strategies and components. More speculative hypothesis require more in depth experimentation and would be potentially misleading with our information base. We therefore focus on insights that can be certainly derived from our data base and provide a solid baseline for future work. Do you think we have been too cautious here and should add some 'bolder' interpretations?
>
> 3) As you mention, the impact of data augmentation is a critical point and we have discussed this as well. Here is our thinking why we have not addressed this dimension more prominently or covered it with more detailed experiments:
>
> - Technical limitation: An informed comparison requires to have similar model checkpoints trained with different data augmentation. With the infrastructure that is available to us we can simply not do this for all frameworks and models in a reasonable time frame. The presented evaluation took several weeks, spread over several months (on RTX 2080 GPUs), blocking also other projects. We could potentially run such ablation studies for a limited subset of models but decided against it for the following reason:
>
> - Not significant enough: We actually have 'comparable' data points for Mask-RCNN R-50 (72 epochs, horizontal flipping augmentation) and R-50 (100 epochs, Large Scale Jitter augmentation), see Figure 10. We find that LSJ augmentation (last SOTA result for Mask R-CNN) has only a minor positive effect on *object-centric* robustness. Importantly, it has a smaller effect than the choice of framework, backbone or neck architecture. Considering the workload of a more in depth comparison we therefore decided against this direction. Note that using stylized images during training would certainly improve robustness but contradicts our objective of finding potential abstractions and generalizations to novel/unseen textures. Regarding the behavior of YOLACT, we agree with your intention but rather see a more in depth analysis as a promising future project. The presented work is already very packed in terms of results.
>
> We still think the reviewer has a valid point here and are willing to include an overview of data augmentation techniques per model in the appendix. Note however that this information is not as well documented as architecture, training schedule or other configuration parameters in the code projects and is sometimes simply not available to us (even on request).
>
> We hope our response addresses most of your concerns an we are looking forward to an ongoing and fruitful discussion. Thanks again for reviewing our paper.

---

> > ### Comment · Reviewer_kn4q · 2021-11-28
> > **response**
> >
> > Thanks for the addressing some of my comments, I think ignoring the data augmentation policies is critical for a fair comparison. Although I understand that there is a technical difficulties in reproducing the numbers while fixing the data augs, it takes a way from the findings of these experiments.

---

> > > ### Author Response · Authors · 2021-11-29
> > > **Response Data Augmentation**
> > >
> > > Thank you for the response.
> > >
> > > Please note that we have a valid comparison in the paper regarding the impact of data augmentation for Mask R-CNN. We find that LSJ data augmentation has indeed a positive but in comparison, only minor positive effect on out-of-distribution robustness. These finding are in line with previous work on out-of-distribution robustness and classification models. See:
> > >
> > > - Hendrycks et al. (2021)  Natural Adversarial Examples CVPR 2021
> > >
> > > https://openaccess.thecvf.com/content/CVPR2021/html/Hendrycks_Natural_Adversarial_Examples_CVPR_2021_paper.html

---

> ### Author Response · Authors · 2021-11-22
> **We uploaded a new version of the paper!**
>
> Hello, we uploaded a new version of the paper. You can find the list of changes here (general comment on top):
>
> https://openreview.net/forum?id=C5Q04gnc4f&noteId=R9ist2EwoJ3

---

### Author Response · Authors · 2021-11-22
**We uploaded a new and reworked version of the paper. Please let us know if you have further questions!**

Hello, we uploaded a substantially reworked version of the paper. In this update, we address the main concerns of all 4 reviews, hopefully to a satisfying extend. Please let us know if you have further questions. We made the following changes:

### Major Improvements

- We tested and report results for 5 additional models (BMask and BCNet). This affects the chapters Model Selection, Results and Discussion.
- We substantially reworked our introduction including the abstract. In particular, we reduce our one-side motivation and discuss related perspectives and real-world challenges to roust instance segmentation. This change also addresses the lack of use case and clarifies our approach, intention and contribution to the community. In addition we address potential biases of our method and discuss its pros and cons as well as alternatives in the related work section.
- We added a representative selection of additional images from Stylized COCO, Objects and Background to the Appendix.

### Minor Improvements

- Updated wording. Consistent use of 'architecture', 'framework' and 'backbone'.
- Updated Figure 6. We removed the very light model lines which makes the figure more clear and improves the viewing performance of the PDF.
- Added a short discussion on the role of data augmentation to the Appendix.
- Reworked results and discussion chapters to support the reader with better interpretations of our findings.

---

### Decision · Program_Chairs · 2022-01-20

**Decision:**

Reject

**Comment:**

This paper finally received divergent and borderline reviews with two positive (6) and two negative (3) rates. After the thorough reviews by ACs ourselves, we would like to decide to reject this work at this time, even though this submission has a lot of potentials including intensive analyses on instance segmentation frameworks and architectures.

We first would like to appreciate comprehensive author’s responses and additional empirical results. They should be extremely helpful to make this submission stronger. Here are some of our suggested points for improvement: (i) The novelty, significance, and practical implications of this work (compared to previous analysis work) may need to be better presented in a more persuasive way. (ii) Nuance of stylization transformation can be better explained compared to other types of perturbations or transformations. (iii) Empirical fairness can be better justified. (iv) Since the paper is written in a highly condensed way, some of reduction may improve the readability. (v) Finally, given that this paper focuses on empirical study about instance segmentation, it may be more appreciated in a computer vision venue.